# In-situ-sprayed therapeutic hydrogel for oxygen-actuated Janus regulation of postsurgical tumor recurrence/metastasis and wound healing

Shuiling Chen [1,2], Yang Luo[1,2], Yang He [1,2], Ming Li [1,2], Yongjian Liu[1,2], Xishen Zhou[1,2], Jianwen Hou [1,2] ✉ & Shaobing Zhou [1,2] ✉

Surgery is the mainstay of treatment modality for malignant melanoma. However, the deteriorative hypoxic microenvironment after surgery is recognized as a stemming cause for tumor recurrence/metastasis and delayed wound healing. Here we design and construct a sprayable therapeutic hydrogel (HIL@Z/P/H) encapsulating tumor-targeted nanodrug and photosynthetic cyanobacteria (PCC 7942) to prevent tumor recurrence/metastasis while promote wound healing. In a postsurgical B16F10 melanoma model in female mice, the nanodrug can disrupt cellular redox homeostasis via the photodynamic therapy-induced cascade reactions within tumor cells. Besides, the photosynthetically generated $O_2$ by PCC 7942 can not only potentiate the oxidative stress-triggered cell death to prevent local recurrence of residual tumor cells, but also block the signaling pathway of hypoxia-inducible factor 1α to inhibit their distant metastasis. Furthermore, the long-lasting $O_2$ supply and PCC 7942-secreted extracellular vesicles can jointly promote angiogenesis and accelerate the wound healing process. Taken together, the developed HIL@Z/P/H capable of preventing tumor recurrence/metastasis while promoting wound healing shows great application potential for postsurgical cancer therapy.

Surgery is a primary therapeutic modality for treating melanoma, which is the most lethal and metastatic malignancy in skin cancer[1–3]. Nevertheless, tumor recurrence and metastasis caused by incomplete surgical resection accounting for >90% cancer deaths remain a huge challenge[4–6]. In addition, the unhealed wound after surgery, which is characterized by large-scale skin defects, usually brings severe postoperative pain and suffering during the recovery phase[7–9]. Therefore, it is a pressing priority to effectively prevent tumor recurrence/metastasis and timely promote wound healing for extending the overall survival and improving the life quality of postoperative patients.

Clinically adjuvant therapies including chemotherapy and radiotherapy are often associated with poor specificity and severe adverse effects[10,11]. Photodynamic therapy (PDT), during which photoactivated photosensitizers convert oxygen ($O_2$) into reactive oxygen species (ROS) to kill tumor cells through causing oxidative damage to cellular macromolecules, shows great promise for specific local tumor ablation due to its noninvasiveness and negligible side effects in healthy tissues[12–15]. However, its therapeutic effect is susceptible to the complex redox homeostasis and adaptation-mediated resistance in tumors[16,17]. Interestingly, nitric oxide (NO), which acts as a critical

[1]Institute of Biomedical Engineering, College of Medicine, Southwest Jiaotong University, Chengdu 610031, China. [2]Key Laboratory of Advanced Technologies of Materials Ministry of Education, School of Materials Science and Engineering, Southwest Jiaotong University, Chengdu 610031, China. ✉e-mail: houjianwen@swjtu.edu.cn; shaobingzhou@swjtu.edu.cn

signaling molecule involved in various physiological processes[18], has been found to effectively disrupt cellular redox homeostasis through accelerating the intracellular glutathione (GSH) catabolism[19]. Moreover, NO exhibits high reactivity with ROS to generate more highly toxic reactive nitrogen species (RNS), which can induce potent nitrosative stress-triggered cell death[20]. While the antitumor efficacy of ROS/NO/RNS are seriously limited by their relatively short half-lives and limited action range[21]. Thus, it is essential to achieve intelligent and spatiotemporal generation of ROS/NO/RNS within tumor cells for maximizing their anticancer efficacy. Hypoxia is one of the most pervasive hallmarks of the tumor microenvironment due to the disbalance between impaired $O_2$ supply and increased $O_2$ demand of rapidly proliferating tumor cells[22–24]. It not only adversely impacts the treatment efficacy of PDT[25,26], but also dramatically activates the expression of hypoxia-inducible factor 1α (HIF-1α) that regulates multiple pivotal steps of tumor metastasis[27–29]. Wound healing is a dynamic and complex process including haemostasis, collagen synthesis, angiogenesis and epithelialization, with which each step is dependent upon an adequate supply of $O_2$[30–32]. While the hypoxic microenvironment of postsurgical wound caused by ischemia seriously delays the wound-healing process[32,33]. Although various methods have been developed to address this issue, it is still a huge challenge to construct a long-lasting $O_2$-supplying system[34–36]. Algal microbes, which are supposed to be the primary $O_2$ suppliers on Earth due to their original photo-energy synthesis system[37–39], could potentially be explored as an desirable oxygenator for alleviating hypoxia.

In this work, we develop a therapeutic hydrogel to prevent tumor recurrence/metastasis and promote wound healing after resection (Fig. 1). To obtain the tumor-targeted nanodrug (denoted as HIL@Z), indocyanine green (ICG) and L-arginine (L-Arg) are loaded into zeolite imidazole framework (ZIF-8) nanoparticle, which is followed by coating with hyaluronic acid (HA)[40] (Fig. 1a). ZIF-8 is chosen as the suitable delivery vehicle due to its high loading capacity, tailored pore size, ease of preparation, and unique pH-responsive biodegradation[41–43]. Then the sprayable calcium alginate hydrogel encapsulating HIL@Z nanodrug and photosynthetic cyanobacteria (PCC 7942) (denoted as HIL@Z/P/H) is constructed in situ at the surgical site (Fig. 1b). After selective internalization and pH-responsive disintegration within tumor cells, the released ICG and L-Arg could produce ROS, NO and RNS under near-infrared (NIR) laser (808 nm) via the PDT-induced cascade reactions, thus disrupting cellular redox homeostasis through simultaneously increasing intracellular reactive species and reducing GSH. More importantly, abundant $O_2$ are continuously produced through photosynthesis of PCC 7942 under Red laser (635 nm) to alleviate the hypoxic microenvironment, which is expected to possess multiple functions, including i) effectively potentiating the PDT-induced nitrosative stress-triggered cell death of residual tumor cells to prevent their local recurrence. ii) significantly blocking the intracellular HIF-1α signaling pathway to inhibit distant tumor metastasis. iii) efficiently upregulating vascular endothelial growth factor (VEGF) with the aid of PCC 7942-secreted extracellular vesicles (EVs), thus promoting angiogenesis and the healing process of postsurgical wound. Therefore, the sprayable HIL@Z/P/H capable of preventing tumor recurrence/metastasis and promoting wound healing holds great promise for postsurgical cancer therapy.

## Results
### Preparation and characterization of HIL@Z nanodrug
HIL@Z nanodrug was prepared through the coordination between $Zn^{2+}$ and 2-methylimidazole (2-MIM) by a simple one-pot self-assembly

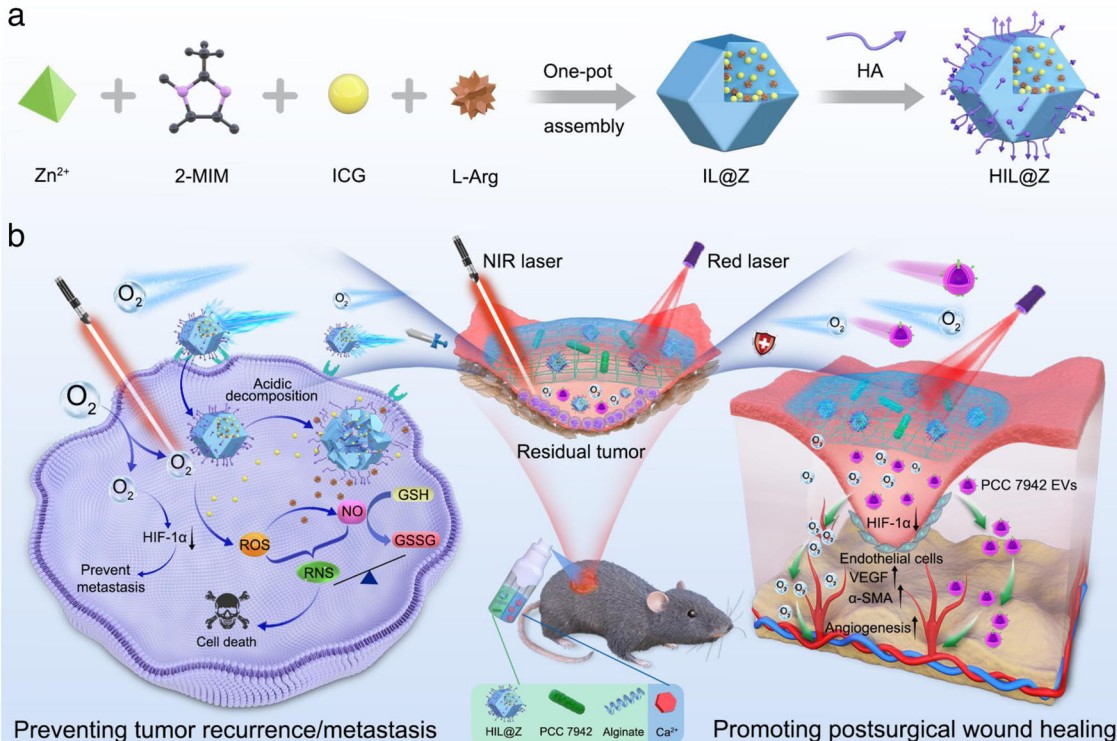

**Fig. 1 | Schematic illustration of sprayable HIL@Z/P/H for efficiently preventing tumor recurrence/metastasis and simultaneously promoting wound healing during the postsurgical cancer treatment. a** Preparation of HIL@Z nanodrug. **b** Schematic showing the in situ formation and action mechanism of sprayed HIL@Z/P/H containing HIL@Z nanodrug and PCC 7942 within the postsurgical wound bed. Under Red laser irradiation, HIL@Z/P/H produces abundant $O_2$ through photosynthesis and effectively relieved the hypoxia microenvironment. In tumor cells, the intracellular cascade reactions induced by HIL@Z nanodrug generate plentiful reactive species (ROS, NO and RNS) and lower the GSH level, accompanied by significant HIF-1α downregulation with the aid of $O_2$, resulting in effective inhibition of residual tumor recurrence/metastasis. Within the postsurgical wound, the excessively generated $O_2$ and PCC 7942-secreted EVs accelerate the wound healing process by downregulating HIF-1α expression and upregulating VEGF level.

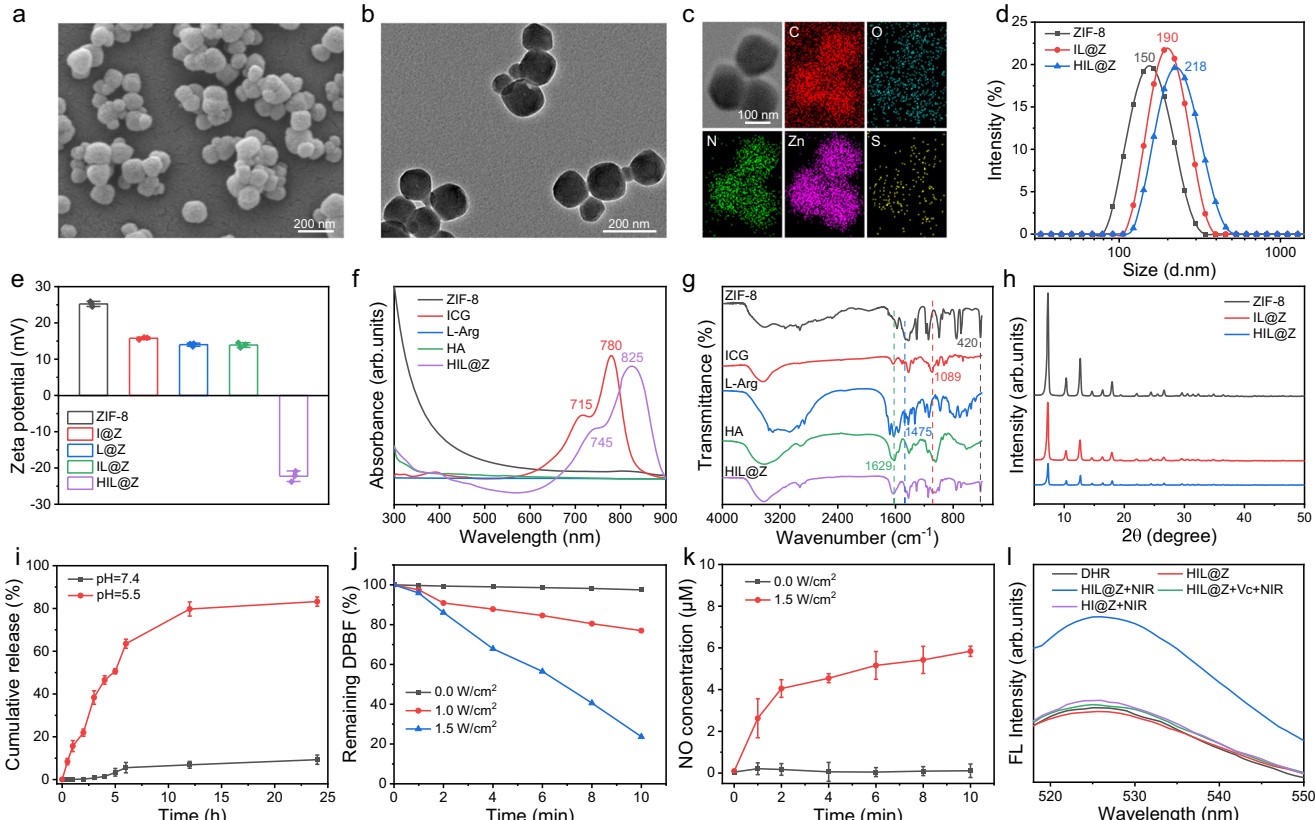

**Fig. 2 | Characterization of nanoparticles. a** SEM, **b** TEM and **c** element mapping images of HIL@Z. **d** Size distribution, **e** zeta potential patterns, **f** UV-vis spectra, **g** FT-IR spectra, and **h** XRD patterns of different nanoparticles. **i** The cumulative release profiles of ICG from HIL@Z in PBS with different pH values. **j** Time-dependent absorbance change of DPBF co-incubated with HIL@Z at 410 nm under NIR irradiation. **k** NO production by HIL@Z with and without NIR irradiation. **l** Fluorescence spectrum of ONOO⁻ characterized by DHR. The results in **a, b** were representative of three independent experiments. Data in **e, i, k** were presented as mean ± SD, $n = 3$ independent samples. Source data are provided as a Source Data file.

strategy to achieve the in situ encapsulation of ICG and L-Arg into ZIF-8 followed by coating hyaluronic acid (HA)[44,45]. The combination of ICG and L-Arg was chosen since it could potentially disrupt the cellular redox homeostasis of tumor cells. On the one hand, PDT-produced ROS can catalyze L-Arg to generate NO to sensitize PDT by down-regulating the intracellular GSH level. On the other hand, NO exhibits high reactivity with ROS to generate more highly active and toxic RNS, thus inducing potent nitrosative stress-triggered cell death through enhancing oxidative damage of intracellular biomolecules. HA coating could endow the nanoparticles with active-targeting capacity towards cancer cells[45]. This could effectively mitigate side effects to normal cells and overcome the short half-lives of ROS/NO/RNS, thereby greatly enhancing their therapeutic effect on tumor cells. Scanning electron microscopy (SEM) and transmission electron microscopy (TEM) images of ZIF-8, IL@Z and HIL@Z nanoparticles all revealed spherical morphologies (Fig. 2a, b and Supplementary Fig. 1). Moreover, elemental mapping results indicated the uniform distribution of C, O, N, Zn and S in HIL@Z, indicating the successful encapsulation of ICG and L-Arg (Fig. 2c). After encapsulating ICG/L-Arg and coating HA, the average hydrodynamic diameter increased from 150 nm to 190 nm and 218 nm, while the zeta potential decreased from +25.5 mV to +14.2 mV and −22.3 mV, respectively (Fig. 2d, e). It has been reported that Zn²⁺ can respectively coordinate with the sulfonic acid group of ICG and guanidine group of L-Arg. So ICG and L-Arg would partake in the crystallization process of ZIF-8, thus leading to larger particle size[44,46]. The distinct reversal of zeta potential was mainly resulted from the inherent negative charge of HA[47]. The incorporation of ICG was further verified using ultraviolet-visible (UV-vis) spectrometry. As shown in Fig. 2f, the characteristic absorption peaks at 715 nm and

780 nm appeared in the UV-vis spectrum of free ICG. For HIL@Z nanoparticles, the typical peaks were located at 745 nm and 825 nm, respectively. The redshift phenomenon implied the formation of ICG oligomers, which was probably induced by the interactions between ICG and the ZIF-8 skeleton or among the ICG molecules[44]. Compared with the fourier-transform infrared spectroscopy (FT-IR) spectrum of ZIF-8, new absorption bands at 1089 cm⁻¹, 1475 cm⁻¹ and 1629 cm⁻¹ appeared in that of HIL@Z, which corresponded to the vinyl stretches of ICG, the C=N stretching vibration of L-Arg, and the C=O stretch of HA (Fig. 2g)[48,49]. These results further verified the successful loading of ICG/L-Arg and functionaliztion of HA. Besides, the X-ray diffraction (XRD) patterns showed that HIL@Z and pure ZIF-8 had similar crystalline characteristic peaks at 2θ values of 7.28, 10.36, 12.71, and 18.02, demonstrating that the crystallinity of the ZIF-8 hosts was hardly influenced by the encapsuled ICG/L-Arg and the coated HA (Fig. 2h)[50]. Next, the contents of ICG, L-Arg and HA in HIL@Z nanoparticles were estimated to be ~7.4%, ~6.9% and ~17.3% from thermogravimetric analysis (TGA) results (Supplementary Fig. 2). Then the stability of HIL@Z nanoparticles was evaluated. It was shown that the hydrodynamic size and polydispersity index of HIL@Z nanoparticles kept constant in RPMI 1640 medium (containing 10% FBS, pH = 7.4) and phosphate-buffered saline (PBS) (pH = 7.4) after one-week storage, illustrating that HIL@Z possessed good stability (Supplementary Fig. 3). Following, the pH-responsiveness of HIL@Z nanoparticles was investigated. The results showed that HIL@Z maintained spherical structure well at neutral pH of 7.4 while broke into small fragments at mildly acidic pH of 5.5 (Supplementary Fig. 4). And the DLS results displayed that the hydrodynamic size of the nanoparticles kept constant at pH = 7.4 whereas remarkably changed at pH = 5.5, further indicating the

disassembly of HIL@Z structure under acidic condition (Supplementary Fig. 5). Then the ICG release behavior from HIL@Z was determined with the assistance of the plotted standard curve (Supplementary Fig. 6). As displayed in Fig. 2i, 6.8% and 79.8% drugs were respectively released in neutral (pH = 7.4) and acidic (pH = 5.5) solutions after 12 h incubation, indicating the excellent pH-responsiveness of ZIF-8 host.

Following, 1,3-diphenylisobenzofuran (DPBF) was applied to characterize the ROS generation of HIL@Z since its absorption intensity at 410 nm would irreversibly weaken in the presence of $^1O_2$[51]. As shown in Fig. 2j and Supplementary Fig. 7, the absorption intensity of DPBF solution sharply decreased once exposure to 808 nm irradiation. In addition, the higher NIR-irradiation power, the faster decrease in the absorption intensity of DPBF. And approximately 76.5% of DPBF was consumed after 808 nm irradiation (1.5 W/cm$^2$) for 10 min. All these results showed that HIL@Z nanoparticles exhibited good ROS-generating capacity under NIR laser. Then Griess assay was used to check the NO production performance when HIL@Z underwent PDT[52]. The concentration of NO was determined based on the standard curve in Supplementary Fig. 8. It was found that the production of NO was heavily dependent on NIR irradiation, which reached 5.8 μM after NIR laser irradiation (1.5 W/cm$^2$) for 10 min (Fig. 2k). Theoretically, the generated NO could further react with ROS to produce peroxynitrite (ONOO$^-$), which was more cytotoxic than ROS and NO. And dihydrorhodamine 123 (DHR) was applied as a specific ONOO$^-$ probe to assess the ONOO$^-$ production. As shown in Fig. 2l, the fluorescence of DHR hardly changed in the DHR+HIL@Z group, while it obviously increased with the extension of NIR irradiation time (Supplementary Fig. 9). And the introduction of vitamin C (Vc) into the DHR+HIL@Z group resulted in an significant decrease in the fluorescence, which was basically the same as that of DHR alone. This was easy to understand since Vc would rapidly scavenge the generated ONOO$^-$. All these results directly proved the production of ONOO$^-$ via the cascade reaction of ROS and NO.

## Preparation and characterization of HIL@Z/P/H

PCC 7942 cells exhibited rod shape with a diameter of 0.6–1.2 μm in width and 3.0–8.0 μm in length on average, and they showed strong red fluorescence under 558 nm excitation as a result of the rich chlorophyll inside them (Supplementary Fig. 10). It could be found that the turbid culture showed the characteristic green color (inset, Supplementary Fig. 11a), which arised from the concentrated intracellular chlorophyll molecules within the exponential growth period (OD$_{680}$ = 0.8–1.2) (Supplementary Fig. 11a). As shown in Supplementary Fig. 11b, the absorption spectrum of the PCC 7942 displayed three strong peaks at 440, 630 and 681 nm, indicating the existence of Chlorophyll a[53]. And the amount of PCC 7942 was determined with the assistance of the standard curve of PCC 7942 (Supplementary Fig. 11c). Then the O$_2$ production capacity of cyanobacteria under 635 nm laser irradiation was investigated. As shown in Supplementary Fig. 12, the O$_2$ production rate was positively correlated with the laser power density and the concentration of PCC 7942 in a certain range. Based on the above results, 1.0 W/cm$^2$ and 8.6 × 10$^8$/mL were chosen as the optimum laser power density and cyanobacteria concentration.

HIL@Z/P/H was prepared by simultaneously spraying equal volume of CaCl$_2$ solution and alginate solution containing HIL@Z nanodrug and PCC 7942[54]. The digital photograph demonstrated that different hydrogels were successfully prepared evidenced by a vial turnover test (Fig. 3a). And the rheology test results showed the storage modulus (G′) of hydrogel and HIL@Z/P/H were always greater than their loss modulus (G″), further indicating their hydrogel characteristics (Supplementary Fig. 13). In addition, the high-magnification pseudocolor SEM image showed that HIL@Z nanoparticles (blue) and PCC 7942 (green) were homogeneously distributed in the microporous network structure of HIL@Z/P/H (Fig. 3b). And the hydrogel morphology was hardly influenced by PCC 7942 (Supplementary

Fig. 14). The element mapping images displayed the presence of C, O, N, and Zn in the HIL@Z/P/H, among which Zn originated from HIL@Z nanoparticles, and C, O, and N were mainly attributed to the hydrogel (Fig. 3c). Moreover, HIL@Z/P/H displayed pH-responsive release of ICG and an effective photodynamic effect (Supplementary Fig. 15 and Supplementary Fig. 16). All these results further verified successful encapsulation of HIL@Z nanoparticles in HIL@Z/P/H. Meanwhile, the fluorescence images further indicated the successful encapsulation of PCC 7942 into HIL@Z/P/H evidenced by the homogeneous distribution of red autofluorescent PCC 7942 (Fig. 3d).

Following, the bioactivity and photosynthetic stability of PCC 7942 within HIL@Z/P/H were systemically studied. It was shown that there were no obvious differences in the OD$_{680}$ value and appearance of the P/H after storing for different days (Supplementary Fig. 17 and Supplementary Fig. 18). Furthermore, PCC 7942 obtained from P/H after storage for different days all grew well and there was no obvious differences in the colony numbers (Supplementary Fig. 19). In addition, the photosynthetic behavior of HIL@Z/P/H showed almost no apparent deterioration after storing for 15 days (Fig. 3e). And HIL@Z/P/H could still produce the same amount of O$_2$ during the repeated laser switch on and off cycling test (Fig. 3f). All these results demonstrated that PCC 7942 encapsulated in HIL@Z/P/H possessed satisfying stability of photosynthetic O$_2$ generation.

## In vitro cytophagocytosis and cellular oxygenation

To prove the cell-targeting ability of HA, Rhodamine B (Rhm B) acting as the ICG substitute was loaded into ZIF-8 to monitor the cellular uptake behavior of HBL@Z nanoparticles. As shown in Fig. 4a, B16F10 cells treated by HBL@Z nanoparticles exhibited much higher fluorescence intensity than that incubated with BL@Z nanoparticles, indicating the CD44-dependent cellular uptake of HBL@Z nanoparticles. Moreover, a competitive inhibition experiment was conducted by preincubating B16F10 cells with free HA for 1.5 h before adding HBL@Z nanoparticles. The resultant intracellular fluorescence intensity, which was similar to that of BL@Z group, was much lower than the HBL@Z group. This was easy to understand since the CD44 receptors were blocked by excessive free HA. Furthermore, quantitative flow cytometry analysis of intracellular Rhm B signal exhibited similar results, which showed that the uptake amount of HBL@Z was double higher than that of BL@Z (Fig. 4b, c). In contrast, human umbilical vein endothelial cells (HUVECs) treated with HBL@Z exhibited weak red fluorescence and there was little difference in the fluorescence intensity among different treatment groups, showing that little nanoparticles were uptaken by HUVECs due to the lack expression of CD44 on them (Supplementary Fig. 20). These results further verified the important role of HA in mediating cellular uptake of HBL@Z. Then the lysosomal escape behavior of HIL@Z was investigated. As shown in Fig. 4d–g, the red fluorescence of HIL@Z mostly overlapped with the green fluorescence of lysosome after incubation for 2.5 h, while the red fluorescence signals in overlapping region quickly decreased and increased in cytoplasm at 4 h. These results not only revalidated the specific targeting action of HIL@Z nanoparticles, but also manifested their successful escape from lysosome.

To evaluate whether the photosynthetic PCC 7942 could alleviate intracellular hypoxic microenvironment, the O$_2$ production was visualized by an intracellular hypoxia indicator Ru(dpp)$_3$Cl$_2$, whose fluorescence could be quenched by O$_2$[55]. In detail, B16F10 cells were deprived of O$_2$ for 12 h before incubating with P/H, and the intracellular O$_2$ levels were monitored by measuring the fluorescence intensities of Ru(dpp)$_3$Cl$_2$. As shown in Fig. 4h, i, strong red fluorescence was observed in the tumor cells of the control, Red and P/H groups, showing their relatively low pO$_2$ value after O$_2$ deprivation. While the fluorescence intensity dramatically decreased in tumor cells after P/H+Red treatment, indicating the effective in vitro oxygenation. As is well known, hypoxia can effectively activate HIF-1α signaling, which influences multiple steps

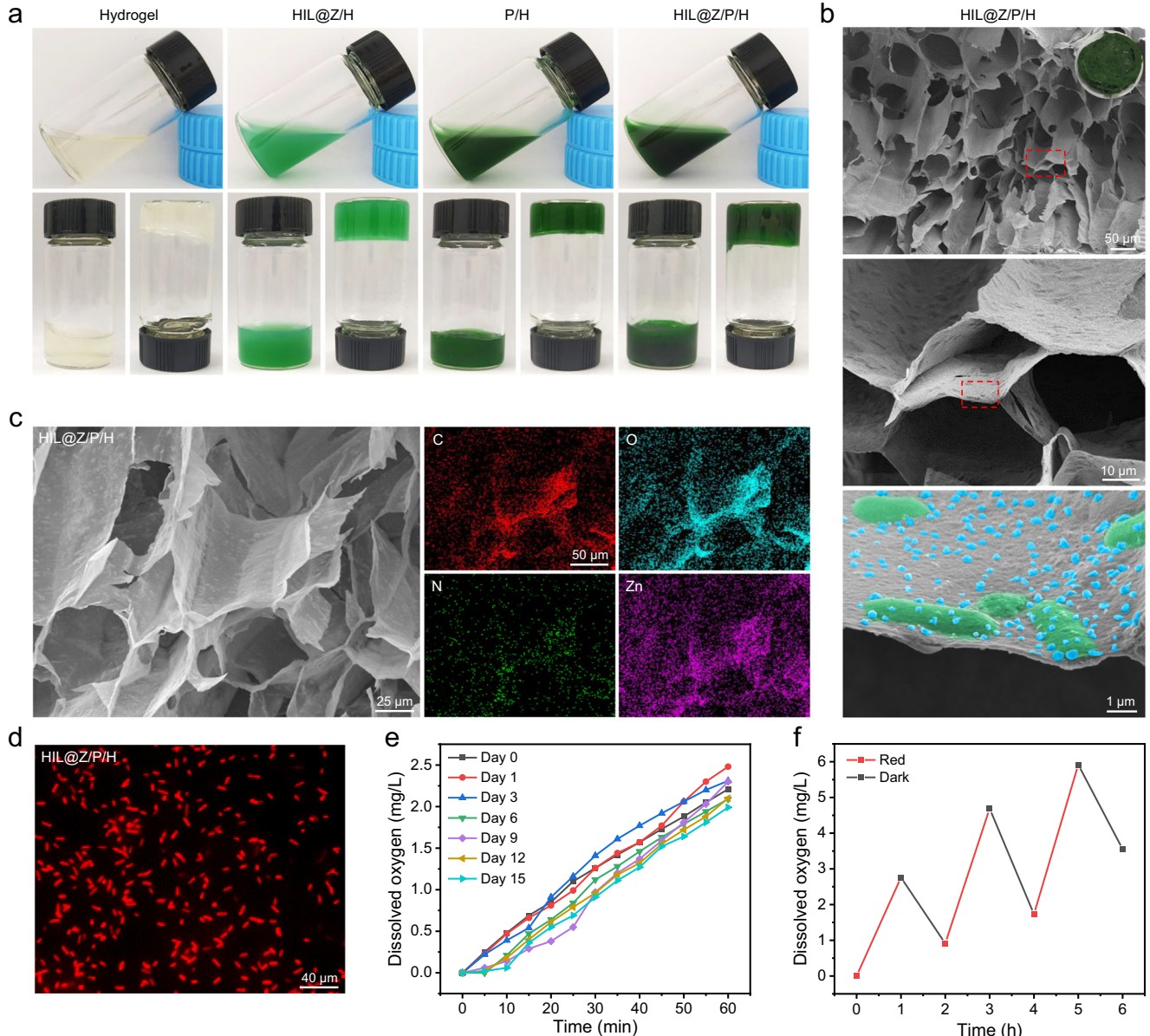

**Fig. 3 | Characterization and photosynthetic O₂-producing capacity of sprayable HIL@Z/P/H. a** Photographs of different hydrogels before and after gelation. **b** SEM images showing the microstructures of HIL@Z/P/H with different magnifications (inset: photograph of the lyophilized HIL@Z/P/H). **c** Element mapping images of HIL@Z/P/H. **d** Fluorescence images of HIL@Z/P/H. **e** Released dissolved O₂ during the storage of HIL@Z/P/H at different days. **f** Light-triggered cyclic O₂ production of HIL@Z/P/H. The results in **b**, **d** were representative of three independent experiments. Source data are provided as a Source Data file.

during tumor metastasis[27]. To verify whether photosynthetic O₂ production of PCC 7942 inhibits the HIF-1α signaling pathway in vitro, WB and RT-qPCR analyses of the cell lysates after different treatments were performed. As shown in Fig. 4j, k, B16F10 cells in the P/H+Red group showed the lowest HIF-1α protein and *HIF-1α* mRNA expression levels among all the groups. Besides, the protein expression levels of HIF-1α-dependent genes, such as MMP-9, EPO, HO-1, ADM and Glut-1, also simultaneously downregulated after hypoxia alleviation (Fig. 4j and Supplementary Fig. 21). All these results demonstrated that the in vitro oxygenation of PCC 7942 could not only alleviate intracellular hypoxic microenvironment, but also effectively block the upstream pathway of HIF-1α-dependent genes closely related to melanoma metastasis.

**In vitro anticancer activity**
The in vitro anticancer activity of HIL@Z/P/H against B16F10 cells was systemically studied. We first evaluated the cytotoxicity of HIL@Z nanoparticles, PCC 7942 and HIL@Z/P/H against HUVECs and B16F10 cells using the alamar blue assay and live/dead staining. The viabilities of HUVECs and B16F10 cells after 24 h incubation was almost higher than 90%, demonstrating all of the samples had good cytocompatibility (Fig. 5a and Supplementary Figs. 22–24). After confirming the good biocompatibility of HIL@Z/P/H, the in vitro anticancer activity of HIL@Z/P/H+Red+NIR irradiation was detected. It was shown that the viability of tumor cells in the HIL@Z/P/H+NIR group decreased to ~33.4%, which was remarkably lower than the other groups (Fig. 5b). Besides, the tumor cells of HIL@Z/P/H+Red+NIR group exhibited the lowest survival rate (19.0%) among all the groups, indicating that the oxygenation effect of PCC 7942 could effectively enhance the cell-killing efficiency. The above results were further demonstrated by live/dead staining fluorescence images (Fig. 5c). The quantitative flow cytometry results (Gating strategy is shown in Supplementary Fig. 25) demonstrated that the total apoptotic ratio of HIL@Z/P/H+NIR group (65.69%)

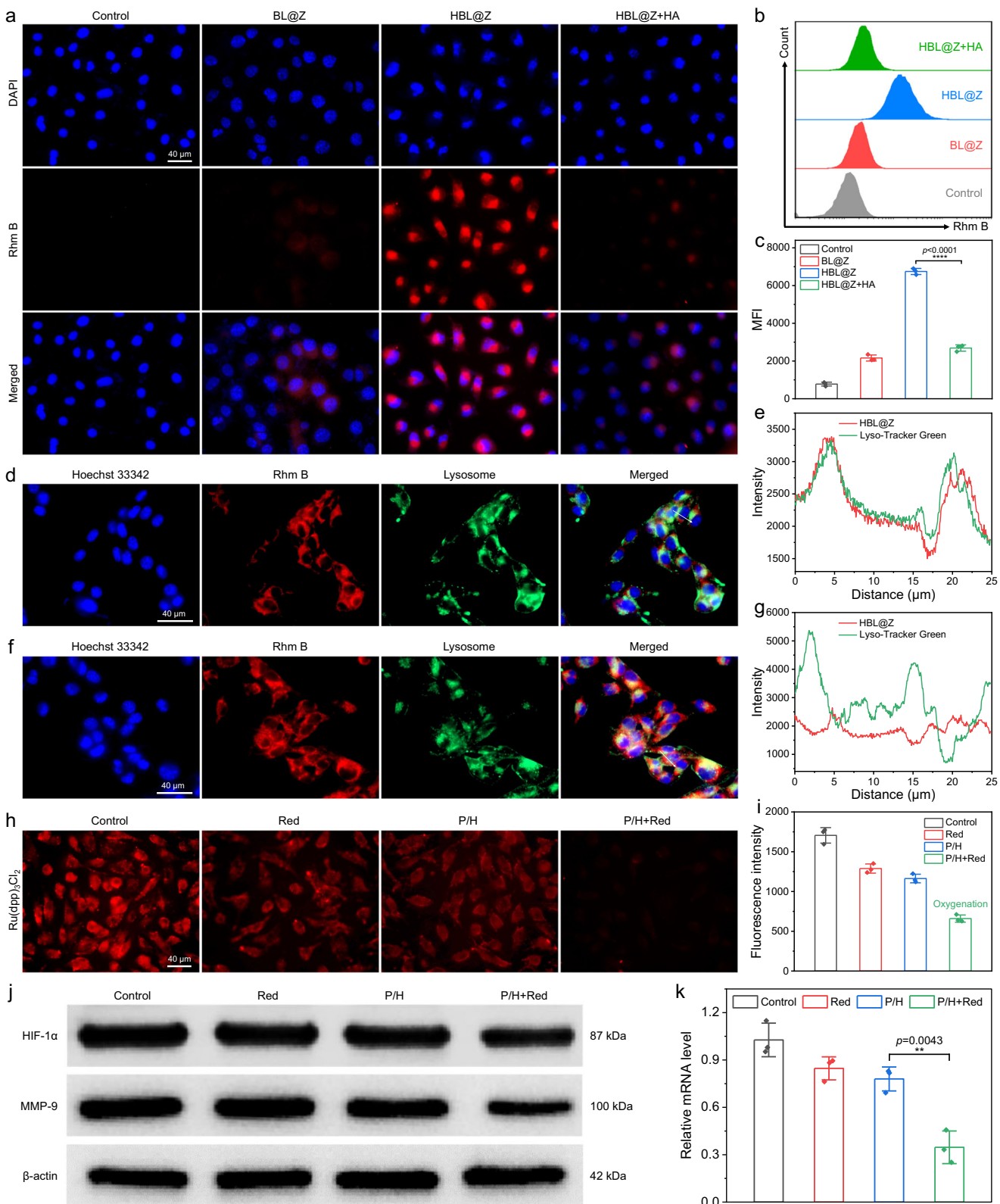

**Fig. 4 | In vitro cytophagocytosis and cellular oxygenation. a** Fluorescence images of B16F10 cells treated with different nanoparticles. **b** Flow cytometry analysis of Rhm B signal and **c** corresponding mean fluorescence intensities (MFI) in B16F10 cells treated with different nanoparticles. Fluorescence images of B16F10 cells incubated with HIL@Z for 2.5 h (**d**) and 4 h (**f**). Blue fluorescence represents the nucleus, red fluorescence represents Rhm B and green fluorescence represents Lyso-Tracker. **e** and **g** are the line scan profiles of the fluorescence intensities at the white arrows in **d** and **f**, respectively. **h** Fluorescence images of Ru(dpp)$_3$Cl$_2$-stained B16F10 cells after different treatments and **i** their corresponding fluorescence intensities. **j** Western blotting (WB) analysis of HIF-1α/MMP-9 protein expressions in B16F10 cells after different treatments. **k** Real-time quantitative polymerase chain reaction (RT-qPCR) analysis of *HIF-1α* mRNA expression in B16F10 cells after different treatments. The results in **a**, **d**, **f** were representative of three independent experiments. Data in **c**, **i**, **k** were presented as mean ± SD, $n = 3$ biologically independent samples. *P* values were calculated via multiple comparisons one-way ANOVA method t-test. Source data are provided as a Source Data file.

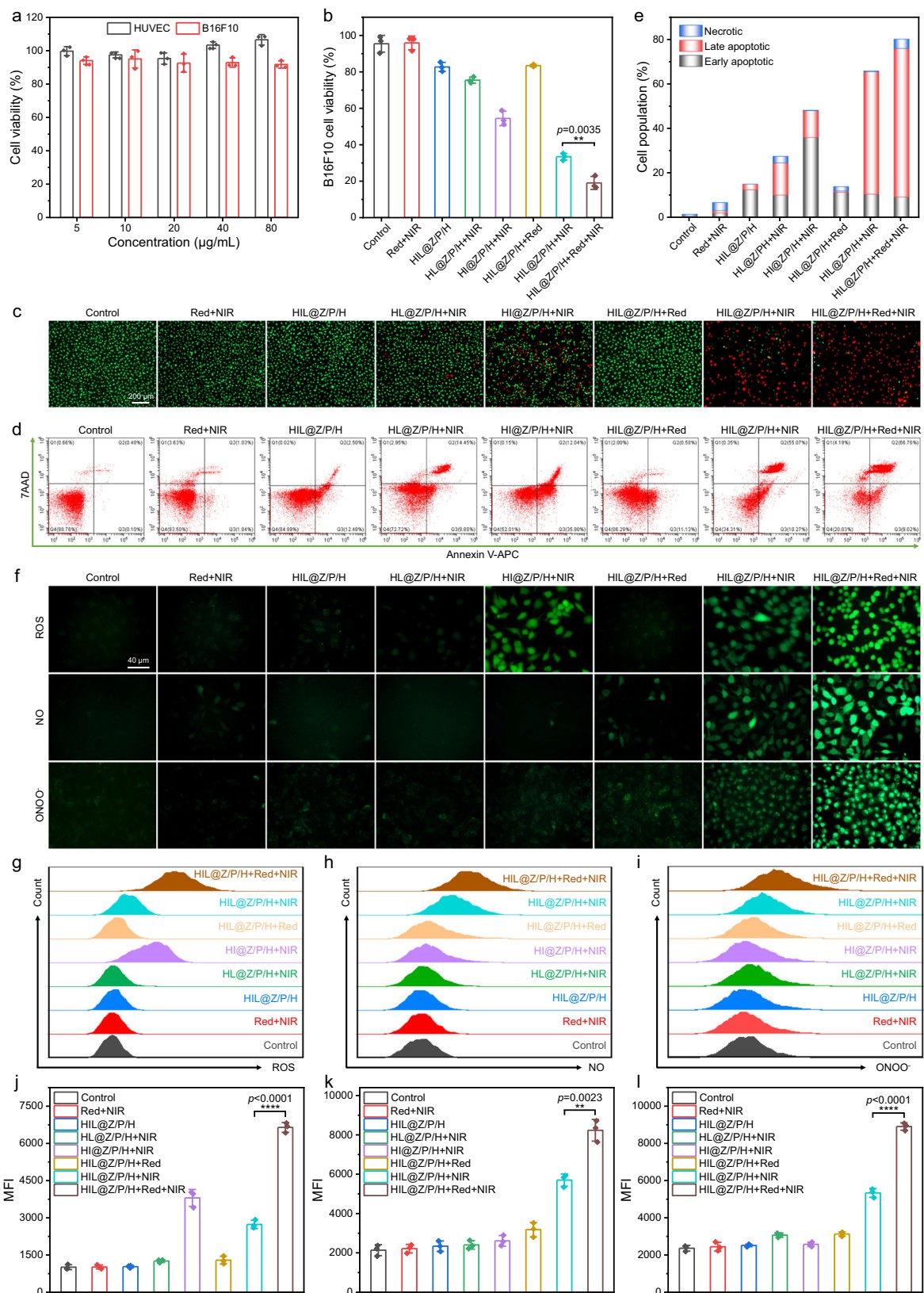

was much higher than that of the control (1.24%) and Red+NIR (6.50%) groups, indicating its excellent apoptosis-inducing feature (Fig. 5d, e). Furthermore, the late apoptotic ratio (66.76%) and the necrotic ratio (4.19%) of the cancer cells treated by HIL@Z/P/H+Red+NIR were conspicuously higher than those (55.07 and 0.35%, respectively) of HIL@Z/P/H+NIR group, further verifying enhanced apoptosis of cancer cells by

the generated O₂. The flow cytometry analysis agreed well with the results of alamar blue assay and live/dead staining, indicating the synergistic effect of ICG/L-Arg and the improvement of photosynthetic generated O₂ on the overall anticancer activity.

To reveal the anticancer mechanism of HIL@Z/P/H+Red+NIR, the intracellular ROS, NO and ONOO⁻ levels in different groups were

**Fig. 5 | In vitro evaluation of the anticancer effect of HIL@Z/P/H on B16F10 cells. a** The relative cell viabilities of HUVECs and B16F10 cells co-cultured with different concentrations of HIL@Z/P/H. **b** The relative cell viabilities of B16F10 cells after different treatments. **c** Fluorescence images of live/dead staining of B16F10 cells in different groups. **d** Flow cytometry analysis of the B16F10 cell apoptosis in different groups. **e** Population of early apoptotic, apoptotic, and necrotic B16F10 cells. **f** Fluorescence images showing intracellular ROS, NO, and RNS detection in B16F10 cells. **g, j** Flow cytometric assay and corresponding MFI of B16F10 cells stained with DCFH-DA (ROS fluorescent probe) after different

treatments. **h, k** Flow cytometric assay and corresponding MFI of B16F10 cells stained with DAF-FM DA (NO fluorescent probe) after different treatments. **i, l** Flow cytometric assay and corresponding MFI of B16F10 cells stained with DHR (ONOO⁻ fluorescent probe) after different treatments. The results in **c**, **f** were representative of three independent experiments. Data in **a**, **b**, **j–l** were presented as mean ± SD, $n = 3$ biologically independent samples. $P$ values were calculated via multiple comparisons one-way ANOVA method t-test. Source data are provided as a Source Data file.

evaluated by 2′,7′-dichlorofluorescein diacetate (DCFH-DA), 3-amino-4-aminomethyl-2′,7′-difluorescein diacetate (DAF-FM DA) and DHR, respectively. As shown in Fig. 5f, the strong green fluorescence of ROS, NO and ONOO⁻ could be observed after B16F10 cells were treated by HIL@Z/P/H+NIR, indicating the effective generation of them inside the cells. Interestingly, the fluorescence intensity of DCFH-DA in the HIL@Z/P/H+NIR-treated cells was a little weaker than that in the HI@Z/P/H+NIR-treated cells. It was easy to understand since the PDT-produced ROS could react with L-Arg to produce NO and ONOO⁻ as discussed above. Remarkably, the fluorescence intensities shown by B16F10 cells in the HIL@Z/P/H+Red+NIR group was more intense than that in the HIL@Z/P/H+NIR group, confirming further increase of ROS, NO and ONOO⁻ levels with the aid of the photosynthetic generated $O_2$. The intracellular generation of ROS, NO and ONOO⁻ was further quantitatively determined by flow cytometry, which was well in agreement with the results of the fluorescence microscope (Fig. 5g–l). In order to determine the GSH-depleting capability of HIL@Z/P/H within cancer cells, ThiolTracker Violet fluorescent dye was used for visualization of intracellular GSH. It was shown that intracellular GSH levels upon HIL@Z/P/H+NIR treatment drastically diminished compared to other groups (Supplementary Fig. 26). And the green fluorescence almost vanished when Red laser irradiation was applied, indicating its excellent capability of scavenging intracellular GSH. All these results revealed that the photosynthetic oxygenation effect of HIL@Z/P/H could effectively promote the ROS/NO/ONOO⁻ production and GSH depletion, thus disrupting redox homeostasis and boosting the anticancer efficacy.

## In vivo tumor recurrence/metastasis inhibition and wound healing promotion in a tumor resection model

Then incomplete tumor resection model was established to further investigate the inhibition effect of HIL@Z/P/H on local recurrence of residual tumor cells. When the tumors grew to a size of about 100 mm³, ~95% tumor was surgically resected and the surgical wound bed was simultaneously sprayed with $CaCl_2$ solution and alginate solution containing HIL@Z nanoparticles and PCC 7942 through a dual-cartridge sprayer. Then Red laser (635 nm, 1.0 W/cm², 30 min) and NIR light (808 nm, 1.5 W/cm², 10 min) were successively applied to the as-prepared HIL@Z/P/H to conduct tumor therapy on days 1, 3 and 5 (Fig. 6a). The wound healing process was recorded and the time-dependent changes in the tumor size were monitored simultaneously during 14 days. It was found that the wound in the HIL@Z/P/H+Red+NIR group gradually closed and even healed on day 14, while the wounds hardly healed with obvious scabs for all other groups (Fig. 6b). In addition, the HIL@Z/P/H+Red+NIR group exhibited superior wound contraction effect and smaller unclosed wound area than the other groups, further verifying its excellent wound-healing-promoting activity (Supplementary Fig. 27). The tumors in the HIL@Z/P/H+Red+NIR group significantly diminished in size, while the tumor growth rates continuously increased uncontrollably in other seven groups. More importantly, some of the tumors in the HIL@Z/P/H+Red+NIR group even disappeared without recurrence (Fig. 6c–e). Moreover, benefiting from the excellent tumor inhibition effect, mice in the HIL@Z/P/H+Red+NIR group showed a markedly prolonged lifespan and displayed the highest survival rate (80%) within 42 days (Fig. 6f).

Furthermore, the in vivo therapeutic effect of HIL@Z/P/H was further investigated by the typical hematoxylin and eosin (H&E) staining, terminal deoxynucleotidyl transferase dUTP nick end labeling (TUNEL), Ki67 staining and HIF-1α immunohistochemical staining (Fig. 6g). The H&E staining results showed most cells in the HIL@Z/P/H+Red+NIR group displayed nuclear rupture and incomplete cell structures, while the tumor tissues of the other seven groups exhibited intact structure with complete cell nuclear. And the TUNEL assay showed that the tumor cells in the HIL@Z/P/H+Red+NIR group exhibited higher apoptotic rate than that in the other seven groups. Besides, the maximum inhibition of Ki67 expression of tumor sections was observed in the HIL@Z/P/H+Red+NIR group, indicating remarkably reduced proliferation of tumor cells. The significant difference among different groups could be further verified by the quantitative analysis (Fig. 6h, i). All these results suggested that HIL@Z/P/H+Red+NIR treatment could effectively suppress the tumor growth by inducing mixed necrosis/apoptosis of tumor cells and inhibiting their proliferation in vivo. To reveal the underlying mechanism, in vivo detection of ROS, NO and RNS were performed. Consistent with in vitro characterization, the fluorescence intensities showing ROS/NO/RNS levels in the HIL@Z/P/H+Red+NIR treatment group were remarkably higher than that in other groups (Supplementary Fig. 28). This further verified that the photosynthetic oxygenation could effectively promote the ROS generation and the successive NO and RNS production, thus potentiating the PDT-induced nitrosative stress-triggered cell death of residual tumor cells to prevent their local recurrence. Furthermore, HIF-1α immunostaining was conducted to evaluate the in vivo hypoxic conditions of tumors in different groups (Fig. 6g). The tumor tissue in the HI@Z/P/H+NIR and HIL@Z/P/H+NIR groups exhibited a mild higher HIF-1α expression than the control group, showing an aggravated hypoxia degree caused by the $O_2$ consumption of PDT. Interestingly, HIF-1α-positive staining (brown) of tumor tissue was hardly observed in the HIL@Z/P/H+Red+NIR group, which was mainly attributed to the $O_2$ supplement to the tumor by PCC 7942. Furthermore, the inhibition capability of $O_2$ to metastasis was further verified by detecting the expression levels of MMP-9 and HIF-1α. As shown in Fig. 6j and Supplementary Fig. 29, the HIL@Z/P/H+Red+NIR treatment dramatically reduced the expression of HIF-1α (by ~60%) and MMP-9 (by ~80%) compared to the control group, indicating the good $O_2$ replenishment and hypoxia-relieving capability of PCC 7942. These in vivo results were well in agreement with the in vitro experiments (Fig. 4j, k), providing strong evidence that the HIL@Z/P/H+Red+NIR treatment maybe alleviate tumor hypoxia to inhibit lung metastasis by downregulating the levels of HIF-1α and MMP-9. Then the metastasis-inhibiting effect of HIL@Z/P/H+Red+NIR treatment was studied by investigating the metastatic nodules in the lung. As shown in Fig. 6k and Supplementary Fig. 30, no noticeable pulmonary metastatic nodules composed of cell masses were present in the mice lungs from the HIL@Z/P/H+Red+NIR group, indicating that HIL@Z/P/H+Red+NIR treatment could effectively suppress tumor metastasis. By contrast, distinct metastatic nodules were found in the mice lungs of the other groups. Besides, the HIL@Z/P/H+NIR group showed several metastatic foci, indicating that $O_2$ rather than reactive species (ROS, NO and RNS) played a key role in inhibiting metastasis in melanoma. It is worth noting that $O_2$ has been found to effectively activate

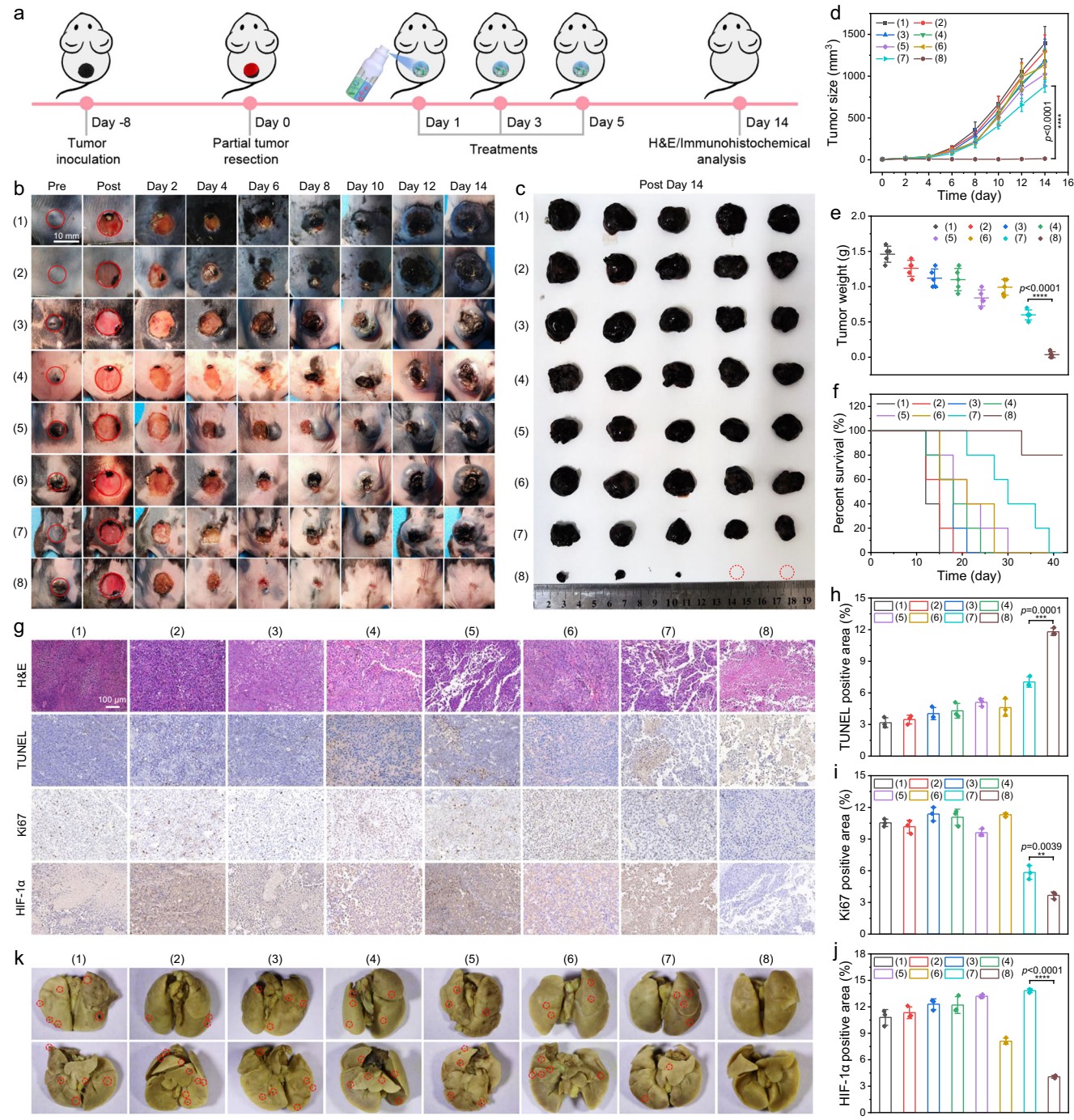

**Fig. 6 | In vivo antitumor performance of HIL@Z/P/H on an incomplete melanoma resection model. a** Schematic illustration of HIL@Z/P/H for inhibiting tumor recurrence in an incomplete melanoma resection model. **b** Photographs of tumor/ wound sites in different groups during the 14-day treatment period. **c** Photographs of the excised tumors after different treatments on day 14. Changes of **d** tumor volume, **e** tumor weight, **f** mouse survival rate, **g** H&E, TUNEL, and Ki67 and HIF-1α stained tumor slices and **h**–**j** their quantification analysis in different groups.

**k** Photographs showing metastatic nodules (red circles) in lung tissues. Treatments: (1) Control, (2) Red+NIR, (3) HIL@Z/P/H (4) HL@Z/P/H+NIR, (5) HI@Z/P/H+NIR, (6) HIL@Z/P/H+Red, (7) HIL@Z/P/H+NIR, (8) HIL@Z/P/H+Red+NIR. The results in **g**, **k** were representative of three independent mice. Data in **d**–**f**, **h**–**j** were presented as mean ± SD, *n* = 3 biologically independent mice in **h**–**j**, *n* = 5 biologically independent mice in **d**–**f**. *P* values were calculated via multiple comparisons one-way ANOVA method *t*-test. Source data are provided as a Source Data file.

antitumor immune responses[56]. We performed immunofluorescence analysis of macrophages and CD8+ T cells infiltrated into the tumors. As shown in Supplementary Fig. 31, the highest percents of F4/80+ and CD3+CD8+ cells were detected in mice treated with HIL@Z/P/H+ Red+NIR, providing the evidence that the changes in the tumor microenvironment, including nitrosative stress-triggered cell death and hypoxia alleviation promoted the infiltration of macrophages and

cytotoxic T cells. Moreover, the body weights of all mice showed insignificant fluctuations during the evaluation period (Supplementary Fig. 32). Furthermore, there were no remarkable differences in hemolysis rate, blood clotting index (BCI), blood biochemistry and blood indictors between the control and treated mice (Supplementary Fig. 33 and Supplementary Fig. 34). And no signs of toxicity in the histological structure of major organ (heart, liver, spleen and kidney)

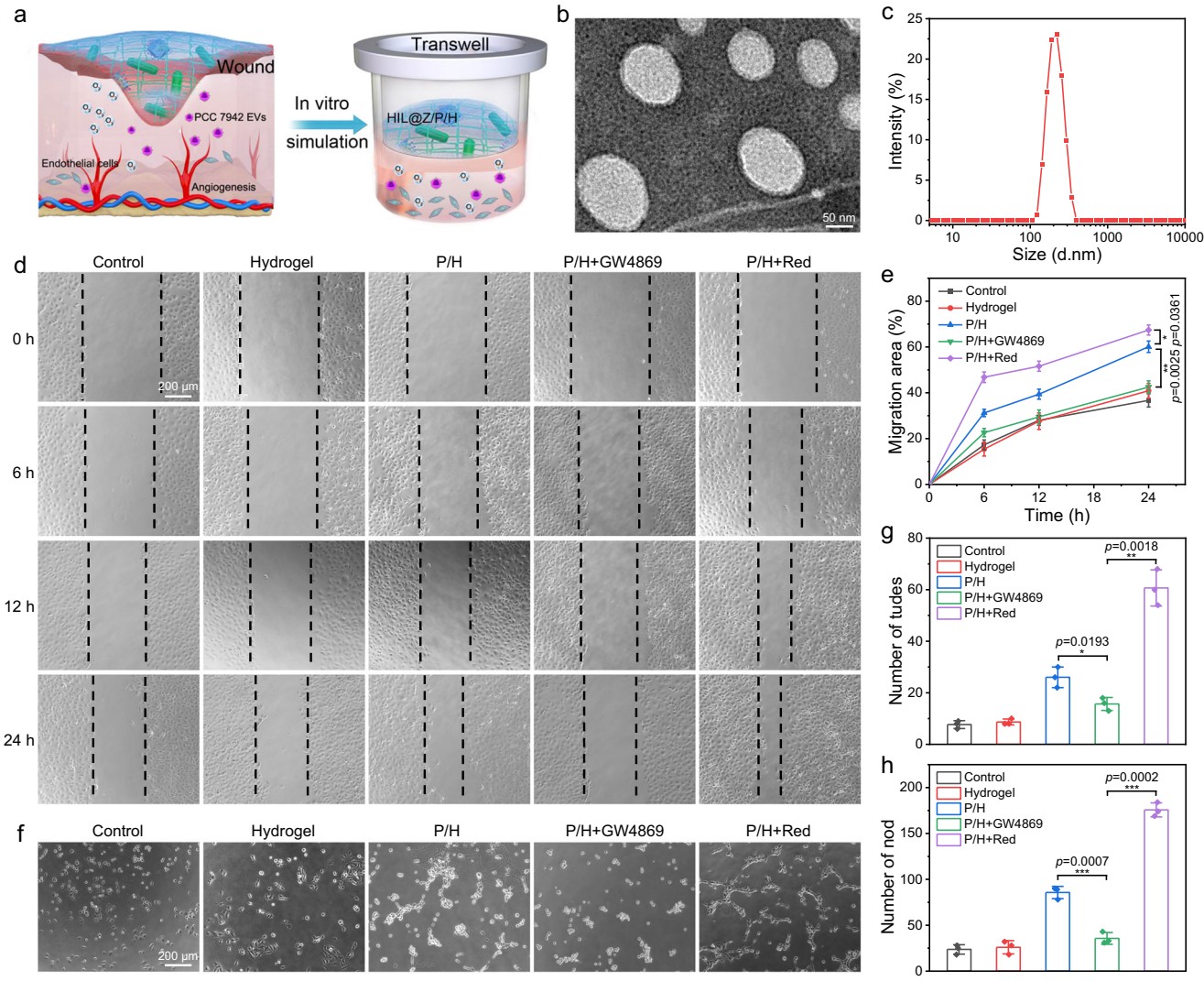

**Fig. 7 | In vitro proangiogenic performance of PCC 7942. a** Illustration showing the wound-healing promotion of P/H. **b** TEM images of PCC 7942-secreted EVs. **c** Hydrated particle size distribution of PCC 7942-secreted EVs. **d, e** Representative images and quantification of HUVEC migration. **f–h** Representative images of HUVECs' tube formation and quantification of tudes and nodes. The results in **b** were representative of three independent experiments. Data in **e**, **g**, **h** were presented as mean ± SD, *n* = 3 biologically independent samples. *P* values were calculated via multiple comparisons one-way ANOVA method t-test. Source data are provided as a Source Data file.

## P/H enhanced angiogenesis in vitro

Angiogenesis is essential in facilitating wound healing by supporting cells with nutrition and $O_2$ through blood supply[33,57]. And long-term hypoxia has been proven to be one of the most potent initiators for angiogenesis. In addition, it has been reported that the PCC 7942-secreted EVs could promote angiogenesis and accelerate cutaneous wound healing by upregulating the interleukin 6 (IL-6) expression[58]. In our study, P/H is expected to effectively supplement sufficient $O_2$ through photosynthesis to reverse hypoxia of the postsurgical wound (Fig. 7a). To validate this, the PCC 7942-secreted EVs were first characterized. As shown in Fig. 7b, c, the PCC 7942-secreted EVs showed a spherical morphology with a diameter of 120–290 nm, which were similar to that previously reported[58]. Then the stimulation effect of P/H+Red treatment on skin tissue regeneration was systemically investigated via cell assays in vitro, such as fibroblast proliferation, wound scratch, and tube formation assay. HUVECs and L929 mouse fibroblasts were chosen as model cells since they played key roles in

vascularization and reepithelialization during the proliferation stage of the wound healing process[59–61]. The alamar blue assays revealed that the viability of HUVECs treated by P/H+Red were higher than the control group (Supplementary Fig. 36). It should be noted that the viability of L929 cells after P/H+Red treatment increased by ~40% than that in the P/H group, indicating that the PCC 7942-produced $O_2$ had notable granulation tissue potential by promoting fibroblast proliferation (Supplementary Fig. 37). All these results demonstrated that the produced $O_2$ had good cell proliferation activity on HUVECs and L929 cells. It has been reported that PCC 7942-secreted EVs also contribute to angiogenesis and wound healing through upregulating the expression of IL-6, which plays an essential role in the process of wound healing by recruiting leukocytes, promoting angiogenesis, and increasing collagen deposition[58,62]. To verify whether the PCC 7942-secreted EVs could upregulate the *IL-6* expression, RT-qPCR analysis was conducted to evaluate *IL-6* mRNA expression in HUVECs and L929 cells. The results showed that the *IL-6* level in the P/H group increased by ~60% than that in the P/H+GW4869 group, further verifying that the upregulation of *IL-6* level maybe account for the promotion activity of PCC 7942-secreted EVs on wound healing (Supplementary Fig. 38).

were identified (Supplementary Fig. 35). All these results suggested HIL@Z/P/H were relatively safe in vivo.

Next, the promotion effect of P/H+Red treatment on cell migration was examined via scratch assay, which was a standard in vitro technique for mimicking the process of wound healing. As shown in Fig. 7d, the scratch in the P/H+Red group almost disappeared after 24 h. And quantitative analysis indicated that the migration ratio of HUVECs in the P/H+Red group (~67.4%) was almost double that of the control group (~36.7%), confirming the photosynthetic generated $O_2$ could effectively accelerate the migration of HUVECs (Fig. 7e). It should be noted that the migration ratio in the P/H+GW4869 group (~42.5%) was obviously lower than that of P/H group (~60.1%), strongly suggesting that the PCC 7942-secreted EVs could effectively enhance the migration of endothelial cell. Furthermore, the vessel-forming performance of HUVECs was evaluated by matrigel tube formation assay. As shown in Fig. 7f–h, the tube number in the control group was ~8, which significantly increased to ~26 in the P/H group and became the highest in the P/H+Red group (~61). Similarly, the number of nodes increased from ~23 (control group) to ~86 (P/H group) and ~176 (P/H+Red group), respectively. Compared with the P/H group, the total numbers of tubes and nodes decreased by ~50% after the addition of GW4869, demonstrating that PCC 7942-secreted EVs endowed the hydrogel with bioactivity to promote vessel forming. Collectively, these results suggested that the $O_2$ and EVs produced by PCC 7942 had good cell angiogenic ability.

### In vivo full-thickness skin defect healing

Based on the animative angiogenesis results in vitro, the promotion of HIL@Z/P/H on wound healing in vivo was further studied (Fig. 8a). To rule out the negative effect of tumor growth on wound healing, a round-shaped full-thickness skin defect (mean-diameter 8 mm) was established on the back of C57BL/6 mice. Followingly, the instant and ultrafast in situ gelation of HIL@Z/P/H (~5 s) was obtained when $CaCl_2$ solution and alginate solution containing HIL@Z nanodrug/PCC 7942 were simultaneously sprayed in the postsurgical cavity (Supplementary Movie 1). And the wound healing process was monitored over 12 days. The intuitive images revealed that the wound healing rates after P/H+Red and HIL@Z/P/H+Red treatments were much higher than other groups (Fig. 8b, c). It could be seen that the wound closure rate in the P/H+Red and HIL@Z/P/H+Red groups even increased to ~62.4% on day 3 post-injury (early phase), which were remarkably higher than other groups. The remaining wound areas on day 12 in the control, hydrogel, P/H, and P/H+GW4869 groups were ~19.3%, ~16.4%, ~6.3%, and ~17.8%, respectively. Whereas the wounds in the P/H+Red and HIL@Z/P/H+Red groups almost completely healed on day 12, and their wound closure rates were up to ~99.3% and ~98.9%. (Fig. 8d, e and Supplementary Fig. 39). In addition, the skin tissues in the PCC/H+Red and HIL@Z/P/H+Red groups were composed of normal architecture full of regular capillary networks, which were totally different from that in the other groups with incomplete vasculature (Supplementary Fig. 40). All these results demonstrated that the $O_2$ and EVs produced by PCC 7942 had satisfactory performance of promoting wound healing by facilitating revascularization.

Histology analyses were conducted to further investigate the wound healing process. It could be seen in the H&E staining results that reconstructed epithelium formed on the edge of the slices in the P/H, P/H+Red and HIL@Z/P/H+Red groups on day 3, while neoepidermis was hardly observed in the wounds of the control, hydrogel and P/H+GW4869 groups on day 7 (Supplementary Fig. 41). On day 12, the wounds in the P/H+Red and HIL@Z/P/H+Red groups showed whole connected and thicker neoepidermis (Fig. 8f, g). Additionally, new hair follicles could be observed in the wounds of P/H+Red and HIL@Z/P/H+Red groups. Furthermore, the collagen depositions of the wounds in the P/H+Red and HIL@Z/P/H+Red groups on days 3, 7 and 12 were always at least two times higher than those in the control and hydrogel groups (Fig. 8h, i and Supplementary Fig. 42). These results demonstrated that HIL@Z/P/H could effectively promote wound healing in vivo by boosting both neoepidermis growth and collagen deposition.

To further reveal the underlying mechanism of enhanced wound healing by PCC 7942-containing hydrogels, immunohistochemical staining of HIF-1α, VEGF, CD31 and α-SMA was carried out to investigate the skin neovascularization. Since hypoxia may imparir the proliferation of repair cells and activation of various growth factors. The immunohistochemistry staining of HIF-1α was firstly determined to evaluate the hypoxia degree in the granulation tissues. It was found that the HIF-1α-positive area in P/H+Red and HIL@Z/P/H+Red groups was about half of that in the other groups, showing the photosynthetically produced $O_2$ by PCC 7942 could relieve hypoxia by supplementing dissolved $O_2$ to the wound bed, thus promoting the healing procedure (Fig. 9a, e and Supplementary Fig. 43). To verify whether hypoxia alleviation could activate various proangiogenic growth factors, VEGF, CD31 and α-SMA which generally represented the level of wound angiogenesis were selected for immunohistochemical staining analysis. It could be seen that the relative VEGF expression and the CD31-positive microvessel densities in the P/H+Red and HIL@Z/P/H+Red groups was about triple that of control and hydrogel groups (Fig. 9b, c, f, g and Supplementary Fig. 44, 45). Similarly, the relative expression level of α-SMA in the P/H+Red and HIL@Z/P/H+Red groups were 6.1% and 5.7%, which was obviously higher than that in the Control (2.7%), Hydrogel (2.6%), P/H (4.3%) and P/H+GW4960 (2.7%) groups (Fig. 9d, h, and Supplementary Fig. 46). In brief, these immunohistochemical staining results suggested that HIL@Z/P/H+Red treatment could effectively alleviate local hypoxia of the wound bed, downregulate the HIF-1α expression and upregulate the expressions of VEGF, CD31 and α-SMA, thus efficiently accelerating wound healing.

## Discussion

Tumor recurrence/metastasis and unhealed wound are two non-negligible issues determining the overall survival and life quality of postsurgical melanoma patients. Hypoxia, a common characteristic of most solid tumors and chronic wounds, is further exacerbated due to the misbalance between impaired $O_2$ supply caused by the seriously damaged microvessels and increased $O_2$ demand of rapidly proliferative tumor cells[23,24,63]. It not only promotes tumor resistance to multiple therapies (PDT, SDT, RT, etc.) by seriously limiting their critical ROS generation, but also dramatically activates the expression of HIF-1α which influences multiple pivotal steps within the metastatic cascade, such as epithelial-mesenchymal transition, invasion and establishment of the premetastatic niche at the distant site and so on[26–29]. Moreover, the deteriorative hypoxia has been found to seriously delay wound healing via impairing angiogenesis, reepithelialization and tissue regeneration, all of which are dependent upon an adequate supply of $O_2$[32,64,65]. Accordingly, various $O_2$-generating systems mainly including "$O_2$-carrying" (hemoglobin, perfluorocarbon, etc.) and "$O_2$-generating" (calcium peroxide, catalase, etc.) strategies have emerged to relieve the hypoxic microenvironment[34,35]. However, the $O_2$ supply of these oxygenation systems can just last for a short time, which can't meet the long-term need of tumor recurrence/metastasis inhibition and wound healing promotion. Therefore, an effective system that can continuously provide $O_2$ to the postsurgical wound is still highly demanded. To address these issues, we developed a portable adjuvant therapeutic system by making full use of the inherent long-lasting $O_2$ self-supplying feature of algal microbes. The abundant photosynthetically generated $O_2$ showed excellent inhibition effect on tumor recurrence/metastasis and prominent promotion action on wound healing. Since $O_2$ has been found to repolarise M2 tumor-associated macrophages to M1 subtype and activate T cells and NK cells[66]. And our study also showed the hypoxia alleviation could effectively promote the infiltration of macrophages/cytotoxic T cells

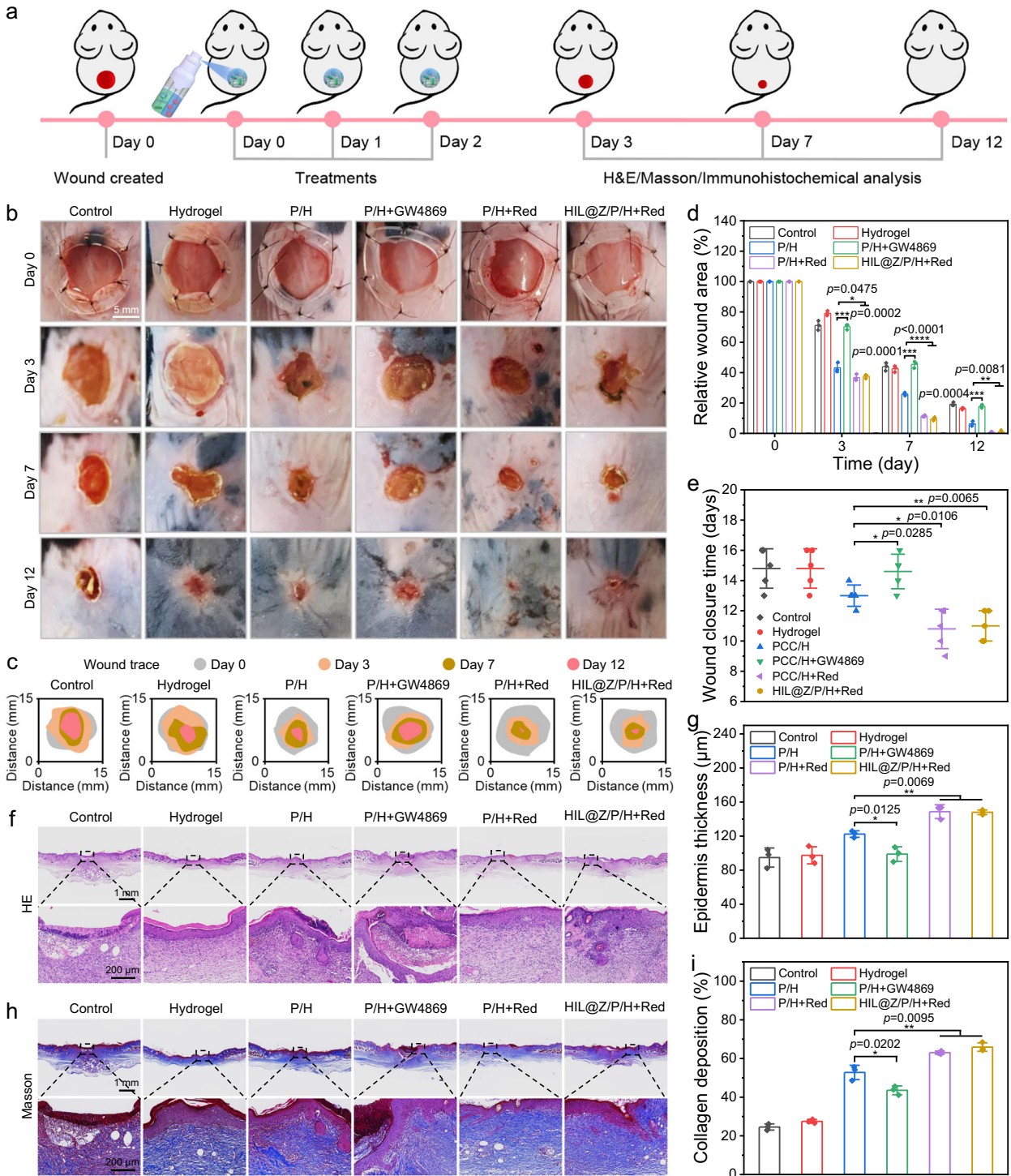

**Fig. 8 | In vivo promotion effect of HIL@Z/P/H on wound healing. a** Schematic illustration of HIL@Z/P/H for promoting wound healing in a mouse model with full-thickness skin defect. **b** Photographs of skin wounds, **c** traces of unhealed wounds, and (**d**) quantitative analysis of the wound areas in different groups during the treatment period. **e** Complete wound closure times in different groups. **f** H&E and **h** Masson staining of the wounds in different groups on day 12. Quantification of the **g** epithelial thickness and **i** collagen deposition in different groups on day 12. Data in **d**, **e**, **g**, **i** were presented as mean ± SD, $n = 3$ biologically independent mice in **d**, **g**, **i**, $n = 5$ biologically independent mice in **e**. $P$ values were calculated via multiple comparisons one-way ANOVA method t-test. Source data are provided as a Source Data file.

into tumors, which maybe has great potential in curing tumors by combining immunotherapy. Overall, it is anticipated that the engineered therapeutic system in this work with long-lasting $O_2$ self-supplying feature is promising in treating various diseases characterized by hypoxia such as cancer, bacterial infections, refractory keratitis, diabetic wounds, ischemic stroke and so on.

In summary, we have developed a HIL@Z/P/H-based therapeutic system to prevent tumor recurrence/metastasis and promote wound healing after resection. Under NIR laser, the tumor-targeted HIL@Z nanodrug could disrupt redox homeostasis by simultaneously increasing intracellular reactive species and reducing GSH via the PDT-induced cascade reactions. And long-lasting and controllable $O_2$

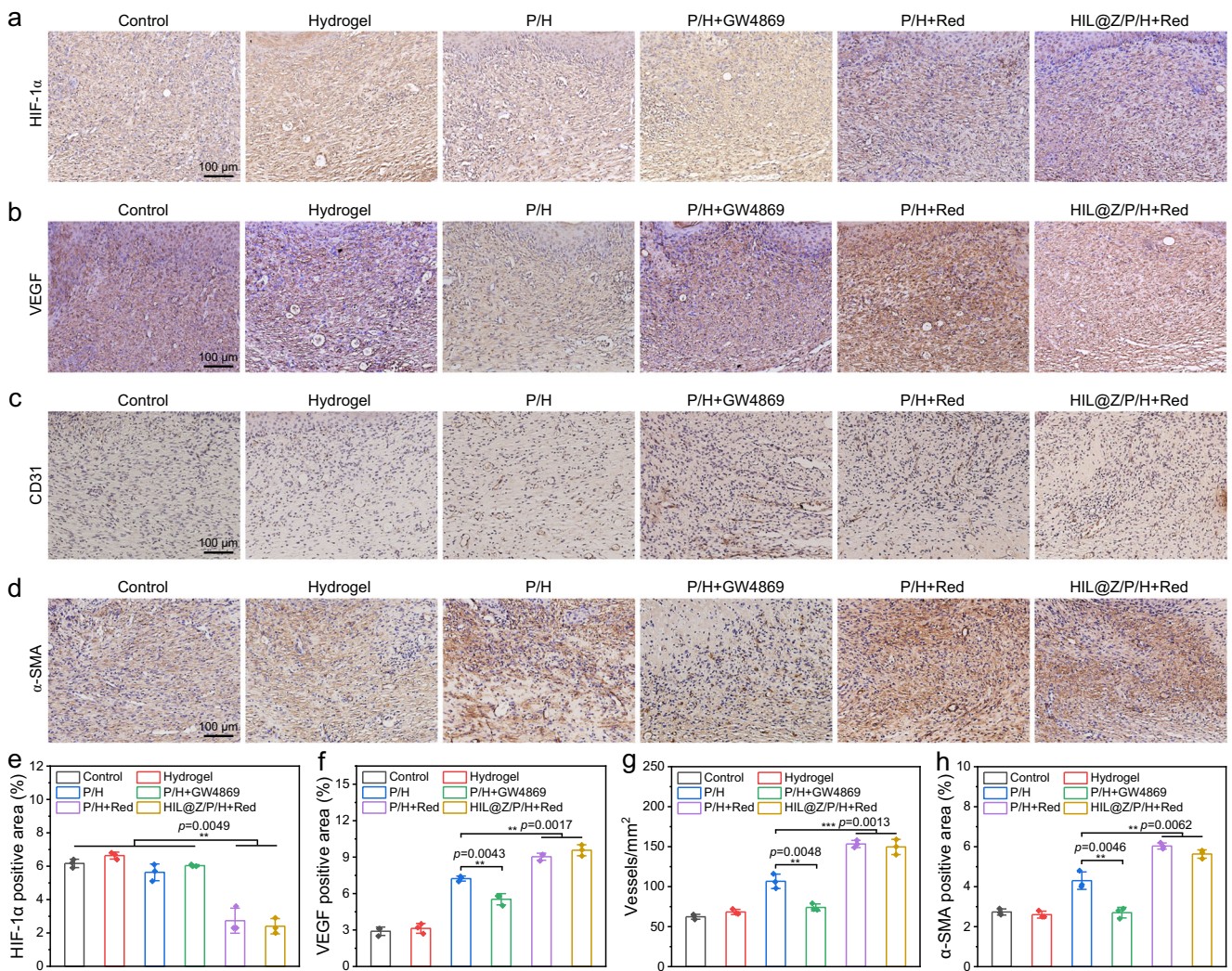

**Fig. 9 | Immunohistological and quantitative analysis of wounds after 12 days of treatment. a–d** HIF-1α, VEGF, CD31 and α-SMA staining of wound tissues after different treatments. **e–h** Quantitative analysis of HIF-1α-positive area, VEGF-positive area, blood vessels and α-SMA-positive area in regenerated dermis after different treatments. Data in **e–h** were presented as mean ± SD, $n = 3$ biologically independent mice. $P$ values were calculated via multiple comparisons one-way ANOVA method t-test. Source data are provided as a Source Data file.

supply produced by the photosynthetic PCC 7942 could effectively reverse the hypoxic microenvironments of both tumor cells and the postsurgical wound bed. The supplemented $O_2$ not only effectively potentiated the PDT-induced nitrosative stress-triggered cell death to prevent local recurrence of residual tumor cells, but also blocked HIF-1α signaling pathway to inhibit their distant metastasis. Furthermore, HIL@Z/P/H exhibited satisfying angiogenesis and wound-healing-promoting behavior ascribed to the synergistic effect of PCC 7942-secreted EVs and photosynthetically generated $O_2$. Taken together, this multifunctional HIL@Z/P/H capable of preventing tumor recurrence/metastasis while promoting wound healing shows great application potential for postsurgical tumor treatment.

## Methods
### Cells and animals
B16F10, HUVEC and L929 cell lines (Cell Bank of Chinese Academy of Sciences, Shanghai, China) were separately cultivated in RPMI 1640 medium (Hyclone), DMEM/F-12 medium (Hyclone) and DMEM medium (Hyclone) at 37 °C in a humidified atmosphere with 5% $CO_2$.

Specific pathogen-free (SPF) C57BL/6 mice (female, 6–8 weeks) were purchased from Chengdu Dossy Experimental Animals Co., Ltd. All mice were housed in a temperature-controlled (25 °C) room under a 12 h dark/12 h light cycle with free access to food and water. And all animal operations were approved by the Institutional Animal Care and Use Committee of Southwest Jiaotong University. The maximum tumor burden permitted was 2000 mm³, and the maximal tumour size/burden in this study was not exceeded.

### Preparation of HIL@Z nanoparticles
The synthesis of IL@Z nanoparticles was performed according to the previous works[44,45]. Typically, ICG (5 mg) and $Zn(NO_3)_2 \cdot 6H_2O$ (0.1 g) were dissolved in ultrapure water (2.4 mL) and stirred (1200 rpm, 5 min) at room temperature. At the same time, L-Arg (5 mg) was added to the aqueous solution of 2-MIM (1.0 g, 4.0 mL) and the mixture was stirred (1200 rpm, 5 min) at room temperature. Then the as-prepared two solutions were mixed and vigorously stirred (1200 rpm, 10 min) at room temperature. The green precipitation was collected by centrifugation (15,300 × g, 10 min) and washed three times using ultrapure water. Similarly, ZIF-8, I@Z and L@Z nanoparticles were prepared using the same approach.

HIL@Z nanoparticles were obtained by dispersing the above as-prepared IL@Z (4 mg/mL) nanoparticles in the HA aqueous solution (2 mg/mL). After stirring (600 rpm) in dark for 24 h, green precipitate was collected by centrifugation (15,300 × g, 30 min) and purified by

ultrapure water to clear away free HA. Similarly, HI@Z and HL@Z nanoparticles were prepared using the same approach.

## PCC 7942 culture and collection

Cyanobacteria *Synechococcus elongatus* PCC 7942 strain was commercially provided by the Freshwater Algae Culture Collection at the Institute of Hydrobiology (FACHB-1061; Chinese Academy of Sciences, Wuhan, China). PCC 7942 were grown in the BG-11 medium under continuous rotation in an incubator (30 °C, 125 rpm). White fluorescence light of 2500–3500 lx was administered in a 12 h dark/12 h light cycle. PCC 7942 were collected by centrifugation (2900 × $g$, 5 min). After discarding the supernatant, the precipitates were washed three times using PBS to obtain PCC 7942 for later use. Then the number of PCC 7942 was determined by manual cell counting and calibration with the $OD_{680}$ in the cuvette. The morphology of PCC 7942 was observed by an optical microscope. The $O_2$-producing capacity of PCC 7942 was measured by the $O_2$ electrodes.

## Isolation and identification of PCC 7942-secreted EVs

PCC 7942-secreted EVs were extracted from PCC 7942 using the differential centrifugation method. Briefly, PCC 7942 were cultured in Erlenmeyer flask containing BG-11 medium. When the optical density reached 0.8–1.0, PCC 7942 and cellular debris were removed by centrifugation (7000 × $g$, 15 min) at 4 °C. Then the supernatant was strained using a 0.45 μm polyvinylidene difluoride (PVDF) filters and ultracentrifuged (200,000 × $g$, 1.5 h) to further remove debris. The purified PCC 7942-secreted EVs suspension was strained through a 0.22-μm-pore filter to avoid any potential contamination. The PCC 7942-secreted EVs were then stored in liquid nitrogen for further use.

## Preparation of sprayable HIL@Z/P/H

The preparation of sprayable HIL@Z/P/H was conducted as follows. First, sodium alginate (200 mg) was dissolved in PBS (10 mL). Then HIL@Z (20 mg) and PCC 7942 (8.6 × $10^9$ cells/mL) were added under stirring (125 rpm). At the same time, $CaCl_2$ (100 mg) was dissolved in ultrapure water (10 mL). Finally, both solutions were stored separately in a small spray bottle for later use. The HIL@Z/P/H was prepared by simultaneously spraying equal volumes of alginate solution (20 mg/mL) containing HIL@Z (2 mg/mL) and PCC 7942 (8.6 × $10^8$ cells/mL) with $CaCl_2$ solution (10 mg/mL). The blank Hydrogel, HI@Z/P/H and HL@Z/P/H nanoparticles were prepared using the same approach.

## Dissolved $O_2$ release test

Dissolved $O_2$ present in 20 mL of PBS was removed by flushing with a nitrogen gas for 30 min and incubated with PCC 7942 or HIL@Z/P/H at 25 °C and irradiated by 635 nm light (0.25, 0.5 and 1.0 W/cm²). The dissolved $O_2$ concentration was measured every 10 min using an $O_2$ probe (Rex, JPBJ-608). To check if the HIL@Z/P/H could generate $O_2$ reversibly, repeated cycles of light were administered. The $O_2$-generating effect of photosynthesis was evaluated by turn the light ON or OFF every 30 min. HIL@Z/P/H was dispersed in PBS in the dark at days 0, 1, 3, 6, 9, 12, and 15. The amount of released $O_2$ was monitored to evaluate the stability of HIL@Z/P/H at different time points.

## In vitro oxygenation detection

The hypoxia-sensitive $Ru(dpp)_3Cl_2$ fluorescence probe was employed to assess the $O_2$-producing capability of P/H when co-cultured with B16F10 cells under 635 nm laser. Typically, B16F10 cells (5 × $10^5$ cells/well) were cultivated in a 6-well plate for 10 h. After cell adherence, the cells were subjected to an anaerobic culture bag with a commercial $O_2$-depriving catalyst for another 12 h. Then, the culture medium was displaced by RPMI 1640 medium containing 100 μL of $Ru(dpp)_3Cl_2$ (500 μg/mL DMSO stock) and the hypoxia condition was maintained for 4 h. And P/H containing PCC 7942 (8.6 × $10^8$ cells/mL) was added. After washing with fresh PBS thoroughly, the cells were irradiated

under 635 nm laser (1.5 W/cm²) for 20 min, followed by another hermetical coincubation for 1 h. Finally, the cells were washed twice and analyzed by fluorescence microscopy.

## Detection of intracellular ROS

The ROS generation inside cells was measured by DCFH-DA (ROS fluorescence probe) assay, which could be oxidized to the highly fluorescent DCF by free radicals. B16F10 cells (5 × $10^5$ cells/well) were cultivated in a 6-well plate for 24 h. Then the cells were subjected to different treatments (Control, Red+NIR, HIL@Z/P/H, HL@Z/P/H+NIR, HI@Z/P/H+NIR, HIL@Z/P/H+Red, HIL@Z/P/H+NIR, HIL@Z/P/H+Red+NIR) for 4 h. During the incubation period, the cells were irradiated with 635 nm (Red) or 808 nm (NIR) laser for 20 min. After discarding the medium, the cells were incubated with DCFH-DA (20 μM) for 30 min. Finally, the cells were washed three times with PBS, analyzed by fluorescence microscopy and assayed by flow cytometry.

## Detection of intracellular NO

The NO generation inside cells was determined by DAF-FM DA (NO fluorescence probe) assay which could exhibit high fluorescenc after reaction with NO. B16F10 cells (5 × $10^5$ cells/well) were inoculated in a 6-well plate. After incubation for 24 h, the cells were subjected to different treatments (Control, Red+NIR, HIL@Z/P/H, HL@Z/P/H+NIR, HI@Z/P/H+NIR, HIL@Z/P/H+Red, HIL@Z/P/H+NIR, HIL@Z/P/H+Red+NIR) for 4 h. During the incubation period, the cells were irradiated by 635 nm or 808 nm laser for 20 min. After washing the cells three times with PBS, cells were incubated with DAF-FM DA (5 μM) for 30 min. Then the cells were washed with PBS three times to remove excess probe. Finally, the cells were analyzed by fluorescence microscopy and assayed by flow cytometry.

## Detection of intracellular $ONOO^-$

The $ONOO^-$ generation inside cells was determined by DHR ($ONOO^-$ fluorescence probe) assay which could react with $ONOO^-$ to exhibit high fluorescence. Typically, B16F10 cells (5 × $10^5$ cells/well) were cultivated in a 6-well plate for 24 h. Then the cells were subjected to different treatments (Control, HIL@Z/P/H, Red+NIR, HL@Z/P/H+NIR, HI@Z/P/H+NIR, HIL@Z/P/H+Red, HIL@Z/P/H+NIR, HIL@Z/P/H+Red+NIR) for 4 h. During the incubation period, the cells were irradiated by 635 nm or 808 nm laser for 20 min. Followingly, the cells were washed with PBS three times and stained by DHR solution (100 μM) for another 30 min. Then the cells were washed with PBS three times to remove excess probe. Finally, the cells were analyzed by fluorescence microscopy and assayed by flow cytometry.

## In vivo tumor recurrence evaluation

The back of SPF female C57BL/6 mice were subcutaneously inoculated with B16F10 cells (3.0 × $10^6$ cells per mouse) to establish melanoma tumor model. When the tumor volume reached 100 mm³, a circular full-thickness skin defect wound (8 mm) was constructed at the tumor site and ~95% tumor tissue was removed. Then the mice were randomly divided into eight groups: (1) Control, (2) Red+NIR, (3) HIL@Z/P/H (4) HL@Z/P/H+NIR, (5) HI@Z/P/H+NIR, (6) HIL@Z/P/H+Red, (7) HIL@Z/P/H+NIR, (8) HIL@Z/P/H+Red+NIR. Starting from day 0 after surgery, a transparent and impermeable membrane dressing (Tegaderm™, 3 M) was used to cover on the hydrogels affixed to the wound, which could form a sealing system between the wound and the dressing, ensuring the unidirectional diffusion of the generated $O_2$ into the postoperative wound while not outside the wound. The mice in Red+NIR, HL@Z/P/H+NIR, HI@Z/P/H+NIR, HIL@Z/P/H+Red, HIL@Z/P/H+NIR and HIL@Z/P/H+Red+NIR groups were irradiated by laser (20 min) for three consecutive days. The body weight and tumor volume of mice were recorded every 2 days within 14 days. The tumor volume (mm³) was calculated as (tumor width)² × (tumor length)/2. The wounds on the back of the C57BL/6 mice were photographed every 2 days within

14 days. Animals were sacrificed after14 days, and the tissues (kidney, liver, heart, spleen, lung, and tumor) were taken out for histological analysis including H&E staining, TUNEL staining, immunohistochemical staining assay, ROS staining, NO staining and RNS staining. Meanwhile, the mice blood was used for blood routine analysis.

### In vivo wound healing evaluation

The circular full-thickness skin defect wound model was established by using SPF female C57BL/6 mice. In brief, the mice were anaesthetized, depilated, disinfected and a full-thickness wound was established on the dorsum center using an 8 mm biopsy punch. A silicone ring (8 mm in diameter) was sewn around the wound to hinder the skin from shrinking. The mice were randomly assigned to six groups ($n = 5$): (1) Control, (2) Hydrogel, (3) P/H, (4) P/H+GW4869, (5) P/H+Red and (6) HIL@Z/P/H+Red. The P/H+Red and HIL@Z/P/H+Red groups were irradiated by laser (30 min) for three successive days. Over the next two weeks, the healing process of the wound was recorded by taking the pictures of the mice on days 0, 3, 7, and 12, and the wound areas were measured using Image J software to assess the healing efficiency. The wound tissues were sliced on day 12 for the pathology wound healing analysis using H&E and Masson's trichrome. In addition, immunofluorescence staining (HIF-1α, VEGF, CD31 and α-SMA) were used to further evaluate the regenerated skin tissue.

### Reporting summary

Further information on research design is available in the Nature Portfolio Reporting Summary linked to this article.

## Data availability

The data that support the findings of this study are available within the article and its Supplementary Information files. Data generated in this study are provided in the Source Data file. Source data are provided with this paper.

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

## Acknowledgements

This work was supported by the National Natural Science Foundation of China (51903214 to J.H., 52033007 to S.Z.), Sichuan Science and Technology Program (2022ZYD0069 to J.H.) and the Fundamental Research Funds for the Central Universities (2682023ZTPY049, 2682023ZTPY074 to J.H.). The authors thank Analytical and Testing Center of Southwest Jiaotong University.

## Author contributions

S.C., J.H., and S.Z. conceived and designed the project. S.C. synthesized and characterized the materials. S.C., Yang.L., Y.H., M.L., Y.L., and X.Z. performed all the in vitro and in vivo experiments. S.C. and J.H. discussed the results, interpreted the data, wrote and revised the paper. J.H. and S.Z. supervised the research. All authors read and approved the final version of the manuscript.

## Competing interests

The authors declare no competing interests.
