## [Peer Review File · Nature Communications]

Reviewers' Comments:

Reviewer #1:

Remarks to the Author:

In this study, Chen and coworkers developed a novel adjuvant therapeutic tool based on sprayable nanocomposite hydrogel (HIL@Z/P/H) for postoperative local tumor-wound management. HIL@Z/P/H demonstrated good inhibitory effects on tumor recurrence/metastasis and excellent promotion performance in wound healing. Overall, this work is well conceived and executed. However, some issues should be addressed prior to consideration for publication of this manuscript.

1. It's better to use DLS analysis to monitor the hydrodynamic size changes of HIL@ZIF-8 nanoparticle in neutral and acidic PBS buffer, for better illustrating the pH-responsiveness of HIL@ZIF-8 nanoparticle.
2. The results of Fig. 4A-C are not enough to prove the tumor-targeting effect of HIL@ZIF-8 nanoparticle. In vitro cytophagocytosis of HIL@ZIF-8 nanoparticle by HUVECs should be included as a control group to further demonstrate its selectivity.
3. "NO has been found to effectively disrupt cellular redox homeostasis through accelerating the intracellular glutathione catabolism" (Page 3, Line 49). It is suggested to investigate the intracellular GSH levels.
4. How long does it take to form the sprayed hydrogel at the postoperative wound site? It's better to provide a video to show the in situ formation process of the sprayed hydrogel.
5. Since fibroblast proliferation is essential for the repair process of wound. It is suggested to investigate the effect of the generated O₂ on the proliferation of fibroblasts.
6. Theoretically, the porous structure of HIL@Z/P/H hydrogel allows the bidirectional permeation of the generated O₂. How to ensure that O₂ only diffuse into the postoperative wound while not outside the wound?
7. The characterization analysis of blood compatibility and biosafety was weak. More characterizations such as in vitro experiments (hemolysis and blood clotting index) and in vivo experiments (blood routine and serum biochemical analysis) of materials should be evaluated.
8. Some references about the nanomaterials-based wound healing study can be cited in the revised manuscript, e.g. ACS Nano, 2020, 14, 3299. Nano Today, 2022, 42, 101368. ACS Applied Materials & Interfaces, 2019, 11, 3809

Reviewer #2:

Remarks to the Author:

This article reported a multifunctional HIL@Z/P/H hydrogel by enveloping targeted HIL@ZIF-8 nanodrug and photosynthetic PCC 7942 cyanobacteria at the surgical site to prevent tumor recurrence/metastasis and promote wound healing. The design is quite ingenious by making full use of multiple functions of O₂. In vitro and in vivo studies have been performed to verify the authors' hypothesis. The topic is important in biomedical science and greatly contributes to the field of postsurgical adjuvant therapy. However, there are several points to be resolved as follows:

1. The authors need to clearly illustrate why they select ZIF-8 as drug carrier in Introduction part.
2. What are the reasons for the increase in hydrated size and redshift phenomenon in UV-vis spectrum after encapsulating ICG and L-Arg (Fig. 2D, 2F)? The authors should give more detailed description and discussion.
3. The authors should evaluate the lysosomal escape performance of HIL@ZIF-8 nanoparticle to monitor in vitro drug delivery.
4. How about the viability of PCC 7942 after encapsulation in the hydrogel?
5. The in vitro experiment results could support the tumor-killing hypothesis with some contents, however, it is not unclear whether it is really happened in vivo. Thus, the authors should perform in vivo detection of ROS, NO and RNS.
6. Is there any significant difference in the survival rate of mice among different treatment groups?
7. To better demonstrate the impact of HIL@Z/P/H on the healing process of postsurgical wound, the authors need to investigate the wound healing rate and residual wound area of different treatment groups.

8. What is the action mechanism of PCC 7942-secreted EVs on angiogenesis and wound healing? Please give more discussions.

Reviewer #3:

Remarks to the Author:

In this manuscript, the authors developed sprayable hydrogel with encapsulated cyanobacteria and nanoparticles with photodynamic capability and the ability to increase oxygen supply within an in vivo wound healing mouse model. The hydrogel (HIL@Z/P/H) is based on the zeolite imidazole framework, L-arginine, PCC 7942, and freshwater cyanobacteria, all coated with hyaluronic acid. The author describes the reduction of metastatic melanoma by treating HIL@Z/P/H+Red+NIR while preventing the local recurrence of residual tumors.

1. The acronyms HIL@Z/P/H and HIL@ZIF-8 must be initially defined within the abstract.
2. Detail the method preparation of hydrogel and nanoparticles and reference any applicable previous publications. Define 2-mim. 2-methylimidazole?
3. If HIL@Z/P and HIL@Z/P/H are synonymous terms, please consistently use one throughout the manuscript. The same rule is for the nanoparticles HIL@Z and HIL@ZIF-8.
4. On page 6, line 118, it is essential to include a reference to support the provided protocol. Reference appropriately any part of the work described before and clearly state the novelty.
5. Are any other models suitable for the pH-responsive hydrogel?
6. Check grammatical errors throughout the manuscript and supplement. The phrase "the survival/migration of endothelial cells and the expression of angiogenic growth factor, as photosynthetically generated..." contains a grammatical error.
7. Line 166 introduces a new acronym (Vc). Please define.
8. Figure 2 captions need to be better described. Define terms, indicate peaks, and give concise descriptions of each figure.
9. The introduction is excessively lengthy and requires condensation.
10. Check line 183. Is there a figure missing?
11. The fluorescence intensity shown in Figure 4A does not match the signal in 4B. Is the significance in 4C compared to the control?
12. In Figure 4F, the low HIF-1 α expression is due to decreasing or accelerating its degradation; the authors must test its mRNA expression level.
13. In Figure 4, besides the MMP9, what about other HIF-1 α target genes expression like EPO, VEGF, HO-1, ADM, and Glut-1? The authors need to select some of these to confirm their conclusion. In addition, whether the PCC 7942 could affect the B16 cell's metabolic like glycolysis or oxidative phosphorylation (The so-called "Warburg effect").
14. In Figure 5D, the authors should compare 7AAD vs. Annexin V instead (nothing is found in text on Annexin V or this assay). Also, flow data in D shows an increase in necrotic population in treatment HIL@Z/P/H+Red+NIR; however, the author summarizes in 5E as early apoptotic. Flow data in 5D does not correlate with the summary in 5E. Additionally, in Figure 5F, the fluorescence intensity does not match the fluorescence intensity of the flow data in G, H, and I. How do you explain these differences? The author is better off showing the MFI of all experiments in G, H, I.

John Collins and Richard Loeser's publication on PRDX may suggest more specific probes for metabolic markers.

15. In Figure 6 in vivo results, has the author observed that more immune cells, especially macrophages or CD8 cells, infiltrated the tumors? Also, similar groups evaluated in Figure 5 should be considered in Figure 6, and the control used in Figure 6 (Red+NIR) should be regarded as in Figure 5.

16. Are the migration process and O₂ generation in Figure 7 hindered if Red+NIR is applied?

17. Some suggestions- Maybe the author can add some content in the discussion part: In the future, what other directions could these hydrogels be used, possibly combined with immunotherapy to provide enough oxygen to immune cells or antimicrobial and anti-inflammation?

18. The authors need to add some discussion. Compared to other investigations about hydrogel like PMID: 33556605 and PMID: 35121358, why is their hydrogel so promising?

Point-by-point response to the reviewers' comments

Reviewer #1

Comments:

In this study, Chen and coworkers developed a novel adjuvant therapeutic tool based on sprayable nanocomposite hydrogel (HIL@Z/P/H) for postoperative local tumor-wound management. HIL@Z/P/H demonstrated good inhibitory effects on tumor recurrence/metastasis and excellent promotion performance in wound healing. Overall, this work is well conceived and executed. However, some issues should be addressed prior to consideration for publication of this manuscript.

Response:

We genuinely appreciate your positive feedback and constructive comments. According to these comments, we have revised our manuscript totally. Our response is given in blue text and changes/additions to the Revised Manuscript/Supporting Information are given in red text.

Question 1: It's better to use DLS analysis to monitor the hydrodynamic size changes of HIL@ZIF-8 nanoparticle in neutral and acidic PBS buffer, for better illustrating the pH-responsiveness of HIL@ZIF-8 nanoparticle.

Response: Thanks for the valuable suggestion. The hydrodynamic size changes of HIL@ZIF-8 nanoparticle in neutral and acidic PBS buffer were monitored. And the corresponding results (Supplementary Fig. 5) were added to page 13 in the Revised Supporting Information:

Supplementary Fig. 5 Hydrodynamic particle size distribution of HIL@Z immersed in PBS buffer with pH 7.4 and pH 5.5 for 24 h.

The relevant discussions were added to page 6 in the Revised Manuscript:

And the DLS results displayed that the hydrodynamic size of the nanoparticles kept constant at pH 7.4 whereas remarkably changed at pH 5.5, further indicating the disassembly of HIL@Z structure under acidic condition (Supplementary Fig. 5).

Question 2: The results of Fig. 4A-C are not enough to prove the tumor-targeting effect of HIL@ZIF-8 nanoparticle. In vitro cytophagocytosis of HIL@ZIF-8 nanoparticle by HUVECs should be included as a control group to further demonstrate its selectivity.

Response: Thanks for the reviewer’s advice. To further demonstrate the selectivity of HIL@ZIF-8 nanoparticle to B16F10 cells, in vitro cytophagocytosis of HIL@ZIF-8 nanoparticle by HUVECs were conducted and the corresponding results (Supplementary Fig. 20) were added to page 28 in the Revised Supporting Information:

Supplementary Fig. 20 (A) Fluorescence images of HUVECs treated with different nanoparticles. (B) Flow cytometry analysis of Rhm B signal and (C) corresponding mean fluorescence intensity (MFI) in HUVECs treated with different nanoparticles. Error bars represent mean \pm s. d. ($n = 3$).

The relevant discussions were added to page 11 in the Revised Manuscript:

In contrast, HUVECs treated with HBL@Z exhibited weak red fluorescence and there was little difference in the fluorescence intensity among different treatment groups, showing that little nanoparticles were uptaken by HUVECs due to the lack expression of CD44 on them (Supplementary Fig. 20).

Question 3: “NO has been found to effectively disrupt cellular redox homeostasis through accelerating the intracellular glutathione catabolism” (Page 3, Line 49). It is suggested to investigate the intracellular GSH levels.

Response: Thanks for the constructive suggestion. The intracellular GSH levels were evaluated using ThiolTracker Violet staining. And the corresponding results (Supplementary Fig. 25) were added to page 33 in the Revised Supporting Information:

Supplementary Fig. 25 (A) Fluorescence images showing intracellular GSH detection in B16F10 cells and (B) their corresponding fluorescence intensities. Error bars represent mean \pm s. d. ($n = 3$); $*p < 0.05$, $**p < 0.01$, $***p < 0.001$.

The relevant discussions were added to page 15 in the Revised Manuscript:

In order to determine the GSH-depleting capability of HIL@Z/P/H within cancer cells, ThiolTracker Violet fluorescent dye was used for visualization of intracellular GSH. It was shown that intracellular GSH levels upon HIL@Z/P/H+NIR treatment drastically diminished compared to other groups (Supplementary Fig. 25). And the green fluorescence almost vanished when Red laser irradiation was applied, indicating its excellent capability of scavenging intracellular GSH. All these results revealed that the photosynthetic oxygenation effect of HIL@Z/P/H could effectively promote the ROS/NO/ONOO⁻ production and GSH depletion, thus disrupting redox homeostasis and boosting the anticancer efficacy.

Question 4: How long does it take to form the sprayed hydrogel at the postoperative wound site? It's better to provide a video to show the in situ formation process of the sprayed hydrogel.

Response: Thanks for your valuable suggestion. It takes about 5 seconds to form the sprayed hydrogel at the postoperative wound site. According to the reviewer's suggestion, a video showing the in situ formation process of the sprayed hydrogel at the postoperative wound site was provided as Supplementary Movie 1.

The relevant discussions were added to page 23-24 in the Revised Manuscript:

Followingly, the instant and ultrafast in situ gelation of HIL@Z/P/H (~5 s) was obtained when CaCl₂ solution and alginate solution containing HIL@Z/P were simultaneously sprayed in the postsurgical cavity (Supplementary Movie 1).

Question 5: Since fibroblast proliferation is essential for the repair process of wound. It is suggested to investigate the effect of the generated O₂ on the proliferation of fibroblasts.

Response: We sincerely appreciate this insightful comment. The effect of the generated O₂ on the proliferation of fibroblasts was studied and the corresponding results (Supplementary Fig. 36) were added to page 44 in the Revised Supporting Information:

Supplementary Fig. 36 (A) Live/dead-staining images of L929 cells cultured in different groups. (B) Living cell ratio of L929 cells cultured in different groups. Error bars represent mean ± s. d. (n = 3); **p* < 0.05, ***p* < 0.01, ****p* < 0.001.

The relevant discussions were added to page 21 in the Revised Manuscript:

It should be noted that the viability of L929 cells after P/H+Red treatment increased by ~40% than that in the P/H group, indicating that the PCC7942-produced O₂ had notable granulation tissue potential by promoting fibroblast proliferation (Supplementary Fig. 36).

Question 6: Theoretically, the porous structure of HIL@Z/P/H hydrogel allows the bidirectional permeation of the generated O₂. How to ensure that O₂ only diffuse into the postoperative wound while not outside the wound?

Response: Thanks for your question. When the sprayable HIL@Z/P/H was affixed to the wound, a transparent and impermeable membrane dressing (Tegaderm™, 3 M) was used to cover it to form a sealing system between the dressing and the wound. And this sealing system could effectively ensure the unidirectional diffusion of the generated O₂ into the postoperative wound while not outside the wound.

The relevant contents were added to page 33 in the Revised Manuscript:

Starting from day 0 after surgery, a transparent and impermeable membrane dressing (Tegaderm™, 3 M) was used to cover on the hydrogels affixed to the wound, which could form a sealing system between the dressing and the wound, ensuring the unidirectional diffusion of the generated O₂ into the postoperative wound while not outside the wound.

Question 7: The characterization analysis of blood compatibility and biosafety was weak. More characterizations such as in vitro experiments (hemolysis and blood clotting index) and in vivo experiments (blood routine and serum biochemical analysis) of materials should be evaluated.

Response: Thanks for your valuable suggestions. The blood compatibility and biosafety of materials were evaluated. And the corresponding results (Supplementary Fig. 32 and Supplementary Fig. 33) were added to page 40 and 41 in the Revised Supporting Information:

Supplementary Fig. 32 (A) Hemolytic percent of RBCs incubated with HIL@Z/P/H with various concentrations. Triton X-100 (0.1%) and PBS are used as positive and negative controls, respectively. Inset: images for direct observation of hemolysis. (B) Blood clotting index changes of HIL@Z/P/H with various concentrations. DI water and PBS are used as positive and negative controls, respectively. Inset: images showing the supernatant of whole blood after incubation with HIL@Z/P/H with various concentrations. Error bars represent mean \pm s. d. ($n = 3$).

Supplementary Fig. 33 (A-F) Hematology parameters and (G-L) blood biochemical indexes of mice after different treatments. Mice treated with PBS solution served as control. Data are presented as mean \pm s.d. ($n = 3$).

The relevant discussions were added to page 21 in the Revised Manuscript:

Furthermore, there were no remarkable differences in hemolysis rate, blood

clotting index (BCI), blood biochemistry and blood indicators between the control and treated mice (Supplementary Fig. 32 and Supplementary Fig. 33).

Question 8: Some references about the nanomaterials-based wound healing study can be cited in the revised manuscript, e.g. ACS Nano, 2020, 14, 3299, Nano Today, 2022, 42, 101368, ACS Applied Materials & Interfaces, 2019, 11, 3809.

Response: Thanks for the suggestion. The related works about the nanomaterials-based wound healing study (Nano Today, 2022, 42, 101368; ACS Nano, 2020, 14, 3299; ACS Applied Materials & Interfaces, 2019, 11, 3809) have been cited in the Revised Manuscript.

39 Hu, H. et al. Microalgae-based bioactive hydrogel loaded with quorum sensing inhibitor promotes infected wound healing. *Nano Today* **42**, 101368 (2022).

60 Qiao, Y. et al. Light-activatable synergistic therapy of drug-resistant bacteria-infected cutaneous chronic wounds and nonhealing keratitis by cupriferous hollow nanoshells. *ACS Nano* **14**, 3299-3315 (2020).

61 Qiao, Y. et al. Laser-activatable CuS nanodots to treat multidrug-resistant bacteria and release copper ion to accelerate healing of infected chronic nonhealing wounds. *ACS Appl. Mater. Interfaces* **11**, 3809-3822 (2019).

Reviewer #2

Comments:

This article reported a multifunctional HIL@Z/P/H hydrogel by enveloping targeted HIL@ZIF-8 nanodrug and photosynthetic PCC 7942 cyanobacteria at the surgical site to prevent tumor recurrence/metastasis and promote wound healing. The design is quite ingenious by making full use of multiple functions of O₂. In vitro and in vivo studies have been performed to verify the authors' hypothesis. The topic is important in biomedical science and greatly contributes to the field of postsurgical adjuvant therapy. However, there are several points to be resolved as follows:

Response:

Many thanks for your positive and warm comments on our work. We address all the insightful technique queries in the following that greatly help us to improve the manuscript and clarify some important points. Our response is given in blue text and changes/additions to the Revised Manuscript/Supporting Information are given in red text.

Question 1: The authors need to clearly illustrate why they select ZIF-8 as drug carrier in introduction part.

Response: Thanks for your valuable suggestion. We have illustrated the reason why we select ZIF-8 as drug carrier in the Introduction part (page 3) of the Revised Manuscript:

ZIF-8 was chosen as the suitable delivery vehicle due to its high loading capacity, tailored pore size, ease of preparation, and unique pH-responsive biodegradation⁴¹⁻⁴³.

Question 2: What are the reasons for the increase in hydrated size and redshift phenomenon in UV-vis spectrum after encapsulating ICG and L-Arg (Fig. 2D, 2F)? The authors should give more detailed description and discussion.

Response: Thanks for your valuable suggestions.

(1) It has been reported that Zn²⁺ can coordinate with the sulfonic acid group of

ICG (ACS Applied Materials & Interfaces, 2021, 13, 48433). Moreover, the guanidine group of L-Arg could also form stable complex structures with Zn^{2+} (Dalton Transactions, 2023, 52, 4752). Therefore, ICG and L-Arg will participate in the formation and crystallization process of ZIF-8 due to their strong coordination with Zn^{2+} , resulting in larger particle size.

The corresponding discussions were added to page 5 in the Revised Manuscript:

The size increase may be attributed to the coordination between Zn^{2+} and encapsulated ICG/L-Arg. It has been reported that Zn^{2+} can respectively coordinate with the sulfonic acid group of ICG and guanidine group of L-Arg. So ICG and L-Arg would participate in the crystallization process of ZIF-8, thus resulting in larger particle size^{44,46}.

(2) The obvious redshift in the UV-vis absorption of HIL@Z compared with free ICG implied the formation of ICG oligomers, which was probably induced by the interactions between ICG and the ZIF-8 skeleton or among the ICG molecules (ACS Applied Materials & Interfaces, 2021, 13, 48433).

The corresponding discussions were added to page 6 in the Revised Manuscript:

The redshift phenomenon implied the formation of ICG oligomers, which was probably induced by the interactions between ICG and the ZIF-8 skeleton or among the ICG molecules⁴⁴.

Question 3: The authors should evaluate the lysosomal escape performance of HIL@ZIF-8 nanoparticle to monitor in vitro drug delivery.

Response: Thanks for the insightful suggestion. The lysosomal escape performance of HIL@ZIF-8 has been monitored and the corresponding results (Fig. R1) were supplemented to Fig. 4 as Fig. 4D-G on page 12 in the Revised Manuscript:

Fig. R1 Fluorescence images of B16F10 cells incubated with HIL@Z for (A) 2.5 h and (C) 4 h. Blue fluorescence represents the nucleus, red fluorescence represents Rhm B and green fluorescence represents Lyso-Tracker. (B) and (D) are the line scan profiles of the fluorescence intensities at the white arrows in (A) and (C), respectively.

The corresponding discussions were added to page 11 and 12 in the Revised Manuscript:

Then the lysosomal escape behavior of HIL@Z was investigated. As shown in Fig. 4D-G, the red fluorescence of HIL@Z mostly overlapped with the green fluorescence of lysosome after incubation for 2.5 h, while the red fluorescence signals in overlapping region quickly decreased and increased in cytoplasm at 4 h. These results not only revalidated the specific targeting action of HIL@Z, but also manifested that the nanoparticles could successfully escaped from lysosome.

Question 4: How about the viability of PCC 7942 after encapsulation in the hydrogel?

Response: Thanks for your question. The viability of PCC 7942 encapsulated in the hydrogel after storage for different days was studied. And the corresponding results (Supplementary Fig. 19) were added to page 27 in the Revised Supporting Information:

Supplementary Fig. 19 (A) Photographs of colonies of PCC 7942 encapsulated in the hydrogel after storage for different days. (B) Quantitative statistics of the number of PCC 7942 colonies through standard plate counting assay. Error bars represent mean \pm s. d. ($n = 3$).

The corresponding discussions were added to page 11 in the Revised Manuscript:

Furthermore, PCC 7942 obtained from P/H after storage for different days all grew well and there was no significant difference in the number of colonies (Supplementary Fig. 19).

Question 5: The in vitro experiment results could support the tumor-killing hypothesis with some contents, however, it is not unclear whether it is really happened in vivo. Thus, the authors should perform in vivo detection of ROS, NO and RNS.

Response: We sincerely appreciate the insightful comments of the reviewer. In vivo detection of ROS, NO and RNS were conducted and the corresponding results (Supplementary Fig. 27) were added to page 35 in the Revised Supporting Information:

Supplementary Fig. 27 Representative ROS, NO, and ONOO^- staining images of tumor tissues from mice after different treatments.

The relevant discussions were added to page 19 in the Revised Manuscript:

To reveal the underlying mechanism, *in vivo* detection of ROS, NO and RNS were performed. Consistent with *in vitro* characterization, the fluorescence intensities showing ROS/NO/RNS levels in the HIL@Z/P/H+Red+NIR treatment group were significantly higher than that in other groups (Supplementary Fig. 27). This further verified that the photosynthetic oxygenation could effectively promote the ROS generation and the successive NO and RNS production, thus potentiating the PDT-induced nitrosative stress-triggered cell death of residual tumor cells to prevent their local recurrence.

Question 6: Is there any significant difference in the survival rate of mice among different treatment groups?

Response: We thank the reviewer for the question. The survival rate of mice in different treatment groups was studied and the corresponding results (Fig. R2) were supplemented to Fig. 6 as Fig. 6F on page 18 in the Revised Manuscript:

Fig. R2 Changes of mouse survival rate in different groups.

The relevant discussions were added to page 18 in the Revised Manuscript:

Moreover, benefiting from the excellent tumor inhibition effect, mice in the HIL@Z/P/H+Red+NIR group had a dramatically prolonged lifespan and displayed the highest survival rate (60%) within 42 days (Fig. 6F).

Question 7: To better demonstrate the impact of HIL@Z/P/H on the healing process of postsurgical wound, the authors need to investigate the wound healing rate and residual wound area of different treatment groups.

Response: Thanks for your valuable suggestion. The wound healing rate and residual wound area of different treatment groups were analyzed. The corresponding results (Supplementary Fig. 26) were added to page 34 in the Revised Supporting Information:

Supplementary Fig. 26 (A) Quantitative analysis of dynamic wound contraction process on

the wound bed in different groups. **(B)** Quantitative analysis of the wound area during recovery process in different groups. Treatments: (1) Control, (2) Red+NIR, (3) HIL@Z/P/H, (4) HL@Z/P/H+NIR, (5) HI@Z/P/H+NIR, (6) HIL@Z/P/H+Red, (7) HIL@Z/P/H+NIR, (8) HIL@Z/P/H+Red+NIR. Error bars represent mean \pm s. d. ($n = 3$); * $p < 0.05$, ** $p < 0.01$, *** $p < 0.001$.

The relevant discussions were added to page 17-18 in the Revised Manuscript:

In addition, the HIL@Z/P/H+Red+NIR group exhibited superior wound contraction effects and smaller unclosed wound areas than the other groups, further verifying its excellent wound-healing-promoting activity (Supplementary Fig. 26).

Question 8: What is the action mechanism of PCC 7942-secreted EVs on angiogenesis and wound healing? Please give more discussions.

Response: Thanks for your valuable suggestions. PCC 7942-secreted EVs have been found to promote angiogenesis and enhance wound healing through up-regulating the expression of interleukin-6 (IL-6) (Theranostics, 2019, 9, 2678; Biomedicines, 2020, 8, 101). Since IL-6 plays an essential role in angiogenesis during skin wound healing, the promotion of IL-6 expression by endothelial cells and fibroblasts in the wound sites is thought to be a mechanism by which PCC 7942-secreted EVs augment angiogenesis and accelerate wound healing process. To verify this hypothesis, real-time quantitative polymerase chain reaction (RT-qPCR) analysis was conducted to evaluate the expression of IL-6 mRNA in HUVECs and L929 cells. And the corresponding results (Supplementary Fig. 37) were added to page 45 in the Revised Supporting Information:

Supplementary Fig. 37 RT-qPCR analysis of the expression of IL-6 mRNA in (A) HUVECs and (B) L929 cells after different treatments. Error bars represent mean \pm s. d. ($n = 3$); * $p < 0.05$, ** $p < 0.01$, *** $p < 0.001$.

The relevant discussions were added to page 22 in the Revised Manuscript:

It has been reported that PCC 7942-secreted EVs also contribute to angiogenesis and wound healing through upregulating the expression of interleukin-6 (IL-6), which plays an essential role in the wound-healing process by regulating leukocyte infiltration, angiogenesis, and collagen deposition^{58,62}. To verify whether the PCC 7942-secreted EVs could upregulate the IL-6 expression, RT-qPCR analysis was conducted to evaluate the expression of IL-6 mRNA in HUVECs and L929 cells. The results showed that the IL-6 level in the P/H group increased by ~60% than that in the P/H+GW4869 group, further verifying that the promotion of IL-6 expression may be a mechanism by which PCC 7942-secreted EVs augment angiogenesis and accelerate wound healing process (Supplementary Fig. 37).

Reviewer #3

Comments:

In this manuscript, the authors developed sprayable hydrogel with encapsulated cyanobacteria and nanoparticles with photodynamic capability and the ability to increase oxygen supply within an in vivo wound healing mouse model. The hydrogel (HIL@Z/P/H) is based on the zeolite imidazole framework, L-arginine, PCC 7942, and freshwater cyanobacteria, all coated with hyaluronic acid. The author describes the reduction of metastatic melanoma by treating HIL@Z/P/H+Red+NIR while preventing the local recurrence of residual tumors.

Response:

We appreciate the valuable feedback from the reviewer. The following comments and suggestions have inspired us to conduct more in-depth studies to improve the manuscript. Our response is given in blue text and changes/additions to the Revised Manuscript/Supporting Information are given in red text.

Question 1: The acronyms HIL@Z/P/H and HIL@ZIF-8 must be initially defined within the abstract.

Response: Thanks for your valuable suggestion. The acronyms HIL@Z/P/H and HIL@ZIF-8 have been initially defined within the abstract of Revised Manuscript.

Question 2: Detail the method preparation of hydrogel and nanoparticles and reference any applicable previous publications. Define 2-mim. 2-methylimidazole?

Response: Thanks for your careful reading. The detailed procedures of preparing hydrogel/nanoparticles and the relevant references have been added to the Revised Manuscript. And 2-mim has been initially defined in the Revised Manuscript.

Question 3: If HIL@Z/P and HIL@Z/P/H are synonymous terms, please consistently use one throughout the manuscript. The same rule is for the nanoparticles HIL@Z and HIL@ZIF-8.

Response: We are sorry for our carelessness. HIL@Z/P and HIL@Z/P/H are synonymous terms and we have displaced HIL@Z/P with HIL@Z/P/H throughout the Revised Manuscript. And HIL@ZIF-8 has been displaced with HIL@Z throughout the Revised Manuscript.

Question 4: On page 6, line 118, it is essential to include a reference to support the provided protocol. Reference appropriately any part of the work described before and clearly state the novelty.

Response: Thank you for the constructive suggestions. The related references (ACS Applied Materials & Interfaces, 2021, 13, 48433; Advanced Functional Materials, 2018, 28, 1802830) have been added to the Revised Manuscript to support the provided protocol and state the novelty of our work.

The relevant discussions were added to page 5 in the Revised Manuscript:

HIL@Z nanodrug was prepared through a simple one-pot self-assembly strategy to achieve the in situ encapsulation of ICG and L-Arg into ZIF-8 followed by coating hyaluronic acid (HA)^{44,45}. The combination of ICG and L-Arg was chosen since it could potentially disrupt cellular redox homeostasis of tumor cells. On the one hand, PDT-produced ROS can catalyze L-Arg to generate NO to sensitize PDT by downregulating the intracellular GSH level. On the other hand, NO exhibits high reactivity with ROS to generate more highly active and toxic RNS, thus inducing potent nitrosative stress-triggered cell death through enhancing oxidative damage of intracellular biomolecules. HA coating could endow the nanoparticles with active-targeting capacity towards cancer cells⁴⁵. This could effectively mitigate side effects to normal cells and overcome the short half-lives of ROS/NO/RNS, thereby greatly enhancing their therapeutic effect on tumor cells.

Question 5: Are any other models suitable for the pH-responsive hydrogel?

Response: Thanks for your question. Except for postsurgical cancer therapy, the developed pH-responsive and O₂-supplying HIL@Z/P/H might be explored for treatment of various kinds of disease characterized by acidic and hypoxic

microenvironments such as bacterial infections, refractory keratitis, diabetic wounds and so on.

Question 6: Check grammatical errors throughout the manuscript and supplement. The phrase "the survival/migration of endothelial cells and the expression of angiogenic growth factor, as photosynthetically generated..." contains a grammatical error.

Response: Thanks for your careful reading. We have thoroughly double-checked the manuscript and supporting information. In the revised manuscript and supporting information, the typographical and grammatical errors have been corrected.

Question 7: Line 166 introduces a new acronym (Vc). Please define.

Response: Thanks for your careful reading. The acronym (Vc) has been initially defined in the Revised Manuscript.

Question 8: Figure 2 captions need to be better described. Define terms, indicate peaks, and give concise descriptions of each figure.

Response: Thanks for your good comment. We have defined terms and indicated peaks of Figure 2. And each figure has been succinctly described in the Revised Manuscript.

Question 9: The introduction is excessively lengthy and requires condensation.

Response: We sincerely appreciate the insightful comment of the reviewer. We have tried our best to condense the introduction in the Revised Manuscript.

Question 10: Check line 183. Is there a figure missing?

Response: Thanks for your careful review. The missing figure "(inset, Supplementary Fig. 11A)" has been supplemented in the Revised Manuscript (page 8, line 168).

Question 11: The fluorescence intensity shown in Figure 4A does not match the

signal in 4B. Is the significance in 4C compared to the control?

Response: Thank you so much for this feedback. We apologize for any confusion Figure 4 caused. The abscissa range of Figure 4B is too large to make the difference between all the groups unapparent. This is the reason why the fluorescence intensity shown in Figure 4A appears to be inconsistent with the signal in Figure 4B. As shown in Fig. R3, we have now amended Figure 4B to make it match well with Figure 4A. Frankly speaking, there is no significance in comparing with the control group in Figure 4C. The control group was mainly used as the calibrator to adjust the parameters of flow cytometer and make the fluorescence peak values are within the proper range.

The corresponding results (Fig. R3) were supplemented to Fig. 4 as Fig. 4A-C on page 12 in the Revised Manuscript:

Fig. R3 (A) Fluorescence images of B16F10 cells treated with different nanoparticles. (B) Flow cytometry analysis of Rhm B signal and (C) corresponding mean fluorescence intensity (MFI) in B16F10 cells treated with different nanoparticles. Error bars represent mean \pm s. d. ($n = 3$); *** $p < 0.001$.

Question 12: In Figure 4F, the low HIF-1 α expression is due to decreasing or accelerating its degradation; the authors must test its mRNA expression level.

Response: This advice is quite valuable. The relative HIF-1 α mRNA expression levels were assayed by real-time quantitative polymerase chain reaction (RT-qPCR) and the corresponding results (Fig. R4) were supplemented to Fig. 4 as Fig. 4J on

page 12 in the Revised Manuscript:

Fig. R4 RT-qPCR analysis of the expression of HIF-1 α mRNA in B16F10 cells after different treatments. Error bars represent mean \pm s. d. ($n = 3$); $**p < 0.01$.

The relevant discussions were added to page 13 in the Revised Manuscript:

To verify whether photosynthetic O₂ production of PCC 7942 inhibits the HIF-1 α signaling pathway in vitro, WB and RT-qPCR analyses of the cell lysates after different treatments were performed. As shown in Fig. 4J-L, B16F10 cells in the P/H+Red group showed the lowest HIF-1 α protein and mRNA expression levels among all the groups.

Question 13: In Figure 4, besides the MMP9, what about other HIF-1 α target genes expression like EPO, VEGF, HO-1, ADM, and Glut-1? The authors need to select some of these to confirm their conclusion. In addition, whether the PCC 7942 could affect the B16 cell's metabolic like glycolysis or oxidative phosphorylation (The so-called "Warburg effect").

Response: Thank you for raising these important questions.

(1) The expressions of EPO, HO-1, ADM, and Glut-1 were evaluated by western blotting (WB) and the corresponding results (Supplementary Fig. 21) were added to page 29 in the Revised Supporting Information:

Supplementary Fig. 21 (A) Western blotting analysis and (B) corresponding gray values of the expression of EPO, HO-1, ADM and Glut-1 proteins in B16F10 cells after different treatments. Error bars represent mean \pm s. d. ($n = 3$); * $p < 0.05$, ** $p < 0.01$, *** $p < 0.001$.

The relevant discussions were added to page 13-14 in the Revised Manuscript:

Besides, the protein expression levels of HIF-1 α -dependent genes, such as MMP-9, EPO, HO-1, ADM and Glut-1, also simultaneously downregulated after hypoxia alleviation. Further quantification results displayed that the expression of HIF-1 α and related proteins in the P/H+Red group almost decreased by ~50% in contrast to that in the control group (Fig. 4K, L and Supplementary Fig. 21).

(2) To study the effect of PCC 7942 on the B16 cell's metabolism such as glycolysis or oxidative phosphorylation, two key glycolytic enzymes including hexokinase-II (HK-2) and lactate dehydrogenase A (LDHA), along with the lactate secretion were examined by WB assay. As shown in Fig. R5, the HK-2/LDHA protein expressions and the extracellular level of lactate in the P/H+Red group significantly decreased as compared with the other groups, indicating the reduction of glycolytic activity. In addition, there was a significant increase in the expression level of phosphorylation of adenosine 5'-monophosphate-activated protein kinase (p-AMPK) in the P/H+Red group than other groups, demonstrating that the generated O₂ greatly contributed to enhancing the cellular oxidative phosphorylation.

Fig. R5 (A) WB analysis and (B) corresponding gray values of the expression of HK2, LDHA, p-AMPK and AMPK proteins in B16F10 cells after different treatments. (C) Lactate secretion in the cell culture supernatant derived from B16F10 cells after different treatments. Error bars represent mean \pm s. d. ($n = 3$); * $p < 0.05$, ** $p < 0.01$, *** $p < 0.001$.

Question 14: In Figure 5D, the authors should compare 7AAD vs. Annexin V instead (nothing is found in text on Annexin V or this assay). Also, flow data in D shows an increase in necrotic population in treatment HIL@Z/P/H+Red+NIR; however, the author summarizes in 5E as early apoptotic. Flow data in 5D does not correlate with the summary in 5E. Additionally, in Figure 5F, the fluorescence intensity does not match the fluorescence intensity of the flow data in G, H, and I. How do you explain these differences? The author is better off showing the MFI of all experiments in G, H, I. John Collins and Richard Loeser's publication on PRDX may suggest more specific probes for metabolic markers.

Response: We are very grateful for your careful review and detailed comments.

(1) According to the reviewer's suggestion, we measured the number of cells in early apoptosis (Annexin V⁺, 7-AAD⁻) and late apoptosis/necrosis (Annexin V⁺, 7-AAD⁺) using flow cytometry. As shown in Fig. R6D and E, the total apoptotic ratio of HIL@Z/P/H+NIR group (79.97%) was much higher than that of the control (1.24%) and Red+NIR (6.50%) groups, indicating its excellent apoptosis-inducing feature. Furthermore, the late apoptotic ratio (66.76%) and the necrotic ratio (4.19%) of the cancer cells treated by HIL@Z/P/H+Red+NIR were conspicuously higher than those (55.07 and 0.35%, respectively) of HIL@Z/P/H+NIR group, further verifying the enhanced apoptosis of cancer cells by the generated O₂.

The corresponding results (R6D, E) were supplemented to Fig. 5 as Fig. 5D, E and

the relevant discussions were added to page 14 in the Revised Manuscript:

As shown in Fig. 5D and E, the total apoptotic ratio of HIL@Z/P/H+NIR group (79.97%) was much higher than that of the control (1.24%) and Red+NIR (6.50%) groups, indicating its excellent apoptosis-inducing feature. Furthermore, the late apoptotic ratio (66.76%) and the necrotic ratio (4.19%) of the cancer cells treated by HIL@Z/P/H+Red+NIR were conspicuously higher than those (55.07 and 0.35%, respectively) of HIL@Z/P/H+NIR group, further verifying enhanced apoptosis of cancer cells by the generated O₂.

(2) We apologize for any confusion Figure 5G-I caused. The abscissa ranges of Figure 5G-I are too large to make the difference between all the groups unapparent. This is the reason why the signal in Figure 5F appears to be inconsistent with the fluorescence intensity shown in Figure 5G-I. And we have now amended Figure 5G-I to make it match well with Figure 5F. According to the reviewer's suggestion, the MFI of all experiments in Figure 5G-I was determined and the corresponding results (R6J-L) were supplemented to Fig. 5 as Fig. 5J-L in the Revised Manuscript:

Fig. R6 In vitro evaluation of the anticancer effect of HIL@Z/P/H on B16F10 cells. (A) The relative cell viability of HUVECs and B16F10 cells co-cultured with different concentrations of HIL@Z/P/H. (B) The relative cell viability of B16F10 cells after different treatments. (C) Fluorescence images of live/dead staining of B16F10 cells in different groups.

(D) Flow cytometry analysis of the B16F10 cell apoptosis from cells in different groups. (E) Population of early apoptotic, apoptotic, and necrotic B16F10 cells. (F) Fluorescence images showing intracellular ROS, NO, and RNS detection in B16F10 cells. (G, H) Flow cytometric assay and corresponding MFI of B16F10 cells stained with DCFH-DA (ROS fluorescent probe) after different treatments. (I, J) Flow cytometric assay and corresponding MFI of B16F10 cells stained with DAF-FM DA (NO fluorescent probe) after different treatments. (K, L) Flow cytometric assay and corresponding MFI of B16F10 cells stained with DHR (ONOO⁻ fluorescent probe) after different treatments. Error bars represent mean \pm s. d. ($n = 3$); * $p < 0.05$, ** $p < 0.01$, *** $p < 0.001$.

(3) We have carefully read the paper recommended by the reviewer (Journal of Biological Chemistry, 2016, 291, 6641), and strongly agreed that quantifying the changes of the antioxidants peroxiredoxin-2 (Prx2) and Prx3 by WB analysis would be cogent for exploring the ROS evolution during the treatment. And the results showed that the HIL@Z/P/H+Red+NIR group exhibited significantly elevated PRX2/PRX3 protein expressions than that in the other groups, further indicating the increased oxidative stress within tumor cells (Fig. R7).

Fig. R7 (A) WB analysis and (B) corresponding gray values of the expression of PRX2 and

PRX3 proteins in B16F10 cells after different treatments. Treatments: (1) Control, (2) Red+NIR, (3) HIL@Z/P/H, (4) HL@Z/P/H+NIR, (5) HI@Z/P/H+NIR, (6) HIL@Z/P/H+Red, (7) HIL@Z/P/H+NIR, (8) HIL@Z/P/H+Red+NIR. Error bars represent mean \pm s. d. ($n = 3$); * $p < 0.05$, ** $p < 0.01$, *** $p < 0.001$.

Question 15: In Figure 6 in vivo results, has the author observed that more immune cells, especially macrophages or CD8 cells, infiltrated the tumors? Also, similar groups evaluated in Figure 5 should be considered in Figure 6, and the control used in Figure 6 (Red+NIR) should be regarded as in Figure 5.

Response: Thanks for your careful reading and valuable suggestions.

(1) We performed immunofluorescence analysis of macrophages and CD8⁺ T cells infiltrated in the tumors. As shown in Supplementary Fig. 29, the highest percent of F4/80⁺ and CD3⁺CD8⁺ cells was detected in mice treated with HIL@Z/P/H+Red+NIR, providing the evidence that the changes in the tumor microenvironment including nitrosative stress-triggered cell death and hypoxia alleviation promoted the infiltration of macrophages and cytotoxic T cells.

Supplementary Fig. 29 Immunofluorescence imaging on the infiltration of F4/80⁺ macrophages and CD3⁺CD8⁺ cells in the tumor tissues from different groups.

The corresponding results (Supplementary Fig. 29) were added to page 37 in the Revised Supporting Information and the relevant discussions were added to page 20 in the Revised Manuscript:

It is worth noting that O₂ can effectively activate antitumour immune responses⁵⁶. We performed immunofluorescence analysis of macrophages and CD8⁺ T cells infiltrated in the tumors. As shown in Supplementary Fig. 29 the highest percents of

F4/80⁺ and CD3⁺CD8⁺ cells were detected in mice treated with HIL@Z/P/H+Red+NIR, providing the evidence that the changes in the tumor microenvironment including nitrosative stress-triggered cell death and hypoxia alleviation promoted the infiltration of macrophages and cytotoxic T cells.

(2) As suggested by the reviewer, the similar groups in Fig. 5 (HL@Z/P/H+NIR and HIL@Z/P/H+Red+NIR groups) were added to Fig. 6, and the control group used in Fig. 6 (Red+NIR) was considered as the control group in Fig. 5.

The revised Fig. 5 and 6 were respectively added to page 16 and page 18 in the Revised Manuscript:

Fig. 5 In vitro evaluation of the anticancer effect of HIL@Z/P/H on B16F10 cells. (A)

The relative cell viability of HUVECs and B16F10 cells co-cultured with different concentrations of HIL@Z/P/H. **(B)** The relative cell viability of B16F10 cells after different treatments. **(C)** Fluorescence images of live/dead staining of B16F10 cells in different groups. **(D)** Flow cytometry analysis of the B16F10 cell apoptosis from cells in different groups. **(E)** Population of early apoptotic, apoptotic, and necrotic B16F10 cells. **(F)** Fluorescence images showing intracellular ROS, NO, and RNS detection in B16F10 cells. **(G, H)** Flow cytometric assay and corresponding MFI of B16F10 cells stained with DCFH-DA (ROS fluorescent probe) after different treatments. **(I, J)** Flow cytometric assay and corresponding MFI of B16F10 cells stained with DAF-FM DA (NO fluorescent probe) after different treatments. **(K, L)** Flow cytometric assay and corresponding MFI of B16F10 cells stained with DHR (ONOO⁻ fluorescent probe) after different treatments. Error bars represent mean \pm s. d. ($n = 3$); * $p < 0.05$, ** $p < 0.01$, *** $p < 0.001$.

Fig. 6 In vivo antitumor performance of HIL@Z/P/H on an incomplete melanoma resection model. (A) Schematic illustration of HIL@Z/P/H for inhibiting tumor recurrence in an incomplete melanoma resection model. (B) Photographs of tumor/wound sites in different groups during the 14-day treatment period. (C) Photographs of the excised tumors after different treatments on day 14. Changes of (D) tumor volume, (E) tumor weight, (F) mouse survival rate, (G) H&E, TUNEL, and Ki67 and HIF-1 α stained tumor slices and (H-J) their quantification analysis in different groups. (K) Photographs of lung tissues and metastatic nodules were black (represented by red circles). Treatments: (1) Control, (2) Red+NIR, (3) HIL@Z/P/H (4) HL@Z/P/H+NIR, (5) HI@Z/P/H+NIR, (6) HIL@Z/P/H+Red, (7) HIL@Z/P/H+NIR, (8) HIL@Z/P/H+Red+NIR. Error bars represent mean \pm s. d.; n = 5, * p < 0.05, ** p < 0.01, *** p < 0.001.

The corresponding results (Supplementary Fig. 28, Supplementary Fig. 30 and Supplementary Fig. 31) were added to page 36, 38 and 39 in the Revised Supporting Information:

Supplementary Fig. 28 (A) MMP-9 staining tumor slices and (B) their quantitative analysis in different groups. Error bars represent mean \pm s. d. ($n = 3$); * $p < 0.05$, ** $p < 0.01$, *** $p < 0.001$.

Supplementary Fig. 30 Representative images of H&E-stained lung tissues in different groups.

Supplementary Fig. 31 Changes of mice body weight in different groups during the 14-day treatment period. Error bars represent mean \pm s. d. ($n = 3$).

Question 16: Are the migration process and O₂ generation in Figure 7 hindered if Red+NIR is applied?

Response: Thank you for this question. The migration process of HUVECs under P/H+Red+NIR treatment was studied. The results showed that the migration ratio of HUVECs in the P/H+Red+NIR group (~69.3%) was similar to that in the P/H+Red (~67.4%), suggesting that the introduction of NIR had negligible effect on the migration process and O₂ generation (Fig. R8).

Fig. R8 (A) Representative images and (B) quantification of HUVECs migration. Error bars represent mean ± s. d.; n = 3.

Question 17: Some suggestions-Maybe the author can add some content in the discussion part: In the future, what other directions could these hydrogels be used, possibly combined with immunotherapy to provide enough oxygen to immune cells or antimicrobial and anti-inflammation?

Response: Thank the reviewer for this great suggestion. We have made an outlook for the potential applications of the developed hydrogel and added it in the Discussion section.

The relevant discussions were added to Discussion on page 28 in the Revised Manuscript:

Since O₂ has been found to repolarise M2 tumour-associated macrophages to M1 subtype and activate T cells and NK cells⁶⁶. And our study also showed the hypoxia alleviation could effectively promote the infiltration of macrophages/cytotoxic T cells

into tumors, which maybe has great potential in curing tumors by combining immunotherapy. Overall, it is anticipated that the engineered therapeutic system in this work with long-lasting O₂ self-supplying feature is promising in treating various diseases characterized by hypoxia such as cancer, bacterial infections, refractory keratitis, diabetic wounds, ischemic stroke and so on.

Question 18: The authors need to add some discussion. Compared to other investigations about hydrogel like PMID: 33556605 and PMID: 35121358, why is their hydrogel so promising?

Response: We sincerely appreciate this insightful comment and constructive suggestion. We have made a comparison between our hydrogel and the hydrogels mentioned by the reviewer [Acta Biomaterialia, 2021, 124, 219; Biomaterials, 2022, 282, 121401), and illustrated the advantages of HIL@Z/P/H in the Discussion section.

The relevant discussions were added to Discussion on page 27 and 28 in the Revised Manuscript:

Tumor recurrence/metastasis and unhealed wound are two non-negligible issues determining the overall survival and life quality of postsurgical melanoma patients. Hypoxia, a common characteristic of most solid tumors and chronic wounds, is further exacerbated due to the imbalance between impaired O₂ supply caused by the seriously damaged microvessels and increased O₂ demand of rapidly proliferative tumor cells^{23,24,63}. It not only promotes tumor resistance to multiple therapies (PDT, SDT, RT, etc.) by seriously limiting their critical ROS generation, but also dramatically activates the expression of HIF-1 α which influences multiple pivotal steps within the metastatic cascade, such as epithelial-mesenchymal transition, invasion and establishment of the premetastatic niche at the distant site and so on²⁶⁻²⁹. Moreover, the deteriorative hypoxia has been found to seriously delay wound healing via impairing angiogenesis, reepithelialization and tissue regeneration, all of which are dependent upon an adequate supply of O₂^{32,64,65}. Accordingly, various O₂-generating systems mainly including “O₂-carrying” (hemoglobin, perfluorocarbon, etc.) and “O₂-generating” (calcium peroxide, catalase, etc.) strategies have emerged to

relieve the hypoxic microenvironment^{34,35}. However, the O₂ supply of these oxygenation systems can just last for a short time, which can't meet the long-term need of tumor recurrence/metastasis inhibition and wound healing promotion. Therefore, an effective system that can continuously provide O₂ to the postsurgical wound is still highly demanded. To address these issues, we developed a new adjuvant therapeutic system by making full use of the inherent long-lasting O₂ self-supplying feature of algal microbes. The abundant photosynthetically generated O₂ showed excellent inhibition effect on tumor recurrence/metastasis and outstanding promotion action on wound healing.

Reviewers' Comments:

Reviewer #1:

Remarks to the Author:

The author figured out all of my concerns.

Reviewer #2:

Remarks to the Author:

accept

Reviewer #3:

Remarks to the Author:

Dear Editor,

In this manuscript, Shuiling Chen et al. developed a novel sprayable hydrogel with encapsulated cyanobacteria and nanoparticles with photodynamic capability against hypoxia microenvironment and tumor recurrence. The calcium alginate hydrogel (HIL@Z/P/H) is based on the zeolite imidazole framework, L-arginine, PCC 7942, and freshwater cyanobacteria. Chen et al. demonstrated the disruptive effect of the targeted therapy under photothermal conditions and with increased apoptotic cell death. Overall, HIL@Z/P/H showed an inhibiting effect against metastatic melanoma by supplying oxygen to postsurgical wounds while promoting wound healing.

The authors responded to all my comments, and this version is suitable for publication.

Sincerely,

Edikan Ogunnaike